# Neural Image Compression with a Diffusion-based Decoder

## Abstract

Diffusion probabilistic models have recently achieved remarkable success in generating high quality image and video data. In this work, we build on this class of generative models and introduce a method for lossy compression of high resolution images. The resulting codec, which we call *DIffuson-based Residual Augmentation Codec (DIRAC)*, is the first neural codec to allow smooth traversal of the rate-distortion-perception tradeoff at test time, while obtaining competitive performance with GAN-based methods in perceptual quality. Furthermore, while sampling from diffusion probabilistic models is notoriously expensive, we show that in the compression setting the number of steps can be drastically reduced.

## 1 Introduction

As a new set of generative approaches, denoising diffusion probabilistic models (DDPMs) (Sohl-Dickstein et al., 2015) have recently shown incredible performance in the generation of high-resolution images with high perceptual quality. For example, they have powered large text-to-image models such as DALL-E 2 (Ramesh et al., 2022) and Imagen (Saharia et al., 2022b), which are capable of producing realistic high-resolution images based on arbitrary text prompts. Likewise, diffusion models have demonstrated impressive results on image-to-image tasks such as super-resolution (Saharia et al., 2022c; Ho et al., 2022a), deblurring (Whang et al., 2022) or inpainting (Saharia et al., 2022a), in many cases outperforming generative adversarial networks (GANs) (Dhariwal & Nichol, 2021). Our goal in this work is to leverage these capabilities in the context of learned compression.

Neural codecs, which learn to compress from example data, are typically trained to minimize distortion between an input and a reconstruction, as well as the bitrate used to transmit the data (Theis et al., 2017). However, optimizing for distortion may result in blurry reconstructions, and recent work has identified the importance of perceptual quality. In particular, Mentzer et al. (2020) demonstrated that a GAN-based image codec is able to output reconstructions that look more realistic to a human observer than a distortion-optimized codec, at the cost of some fidelity to the original. This tradeoff between perceptual quality and fidelity is a fundamental one (Blau & Michaeli, 2019), and finding a good operating point is not trivial and likely application-dependent.

Ideally, one selects this operating point at test time. Although adaptive rate control is commonly used, only the work of Iwai et al. (2021) allows to balance distortion and perceptual quality dynamically. To the best of our knowledge, our work proposes the first neural codec that can navigate the full rate-distortion-perception tradeoff at test time. An earlier diffusion-based codec has been proposed by Theis et al. (2022), but their encoding scheme is prohibitively expensive. Concurrent work by Yang & Mandt (2022) demonstrated a more practical diffusion-based codec, but we show that different design choices allow us to outperform their model by a large margin.

Our approach, called *DIffusion Residual Augmentation Codec (DIRAC)*, uses a base codec to produce an initial reconstruction with minimal distortion, then enhances its perceptual quality using a denoising diffusion probabilistic model. By varying the number of sampling steps, the DDPM can smoothly interpolate between a high fidelity output and one with high perceptual quality. Combined with a multi-rate base codec, this enables a user to navigate the axes of rate-distortion and distortion-perception independently using separate control parameters. Moreover, by making a high fidelity initial prediction first, DDPM sampling can be stopped at any point, allowing the codec to operate more efficiently. We show that we can alter the sampling schedule at test time, and that we obtain strong performance with 20 sampling steps or less.

In summary, this paper makes the following contributions:

1. We demonstrate one of the first practical diffusion-based codecs for high-resolution image compression, with competitive performance in both distortion and perceptual quality.

2. We show that this codec enables fine dynamic control over all independent axes of the rate-distortion-perception tradeoff at test time. To the best of our knowledge, this is the first neural codec with this ability.

3. Although DDPMs are notoriously expensive to sample from, we show that our design choices enable a substantial speedup, allowing sampling in 20 steps or fewer.

## 2 METHOD

### 2.1 DENOISING DIFFUSION PROBABILISTIC MODELS

Denoising diffusion probabilistic models (DDPMs) (Sohl-Dickstein et al., 2015; Ho et al., 2020) are latent variable models in which the latent variables $\mathbf{x_1}, ..., \mathbf{x_T}$ are defined as a $T$-step Markov chain with Gaussian transitions. Through the Markov chain, the forward process of DDPMs gradually corrupts the original data $\mathbf{x_0}$:

$$q(\mathbf{x_{1:T}}|\mathbf{x_0}) = \prod_{t=1}^{T} q(\mathbf{x_t}|\mathbf{x_{t-1}}) , \quad q(\mathbf{x_t}|\mathbf{x_{t-1}}) = \mathcal{N}(\mathbf{x_t}; \sqrt{1 - \beta_t}\mathbf{x_{t-1}}, \beta_t\mathbf{I}) . \quad (1)$$

Here, $q(\mathbf{x_t}|\mathbf{x_{t-1}})$ is a Gaussian distribution that models step $\mathbf{x_t}$ conditioned on the previous step $\mathbf{x_{t-1}}$. It is also possible to further condition these transitions on $\mathbf{x_0}$ (Song et al., 2020). The variances $\beta_t$ are typically chosen empirically and referred to as the *noise schedule*, although they can be learned as well (Dhariwal & Nichol, 2021).

The key idea of DDPMs is that, if the corrupted data at step $T$ follows a distribution which permits efficient sampling, we can construct a generative model by reversing the forward process. For example, the series $q(\mathbf{x_t}|\mathbf{x_{t-1}})$ in Eq. (1) converges to a standard normal distribution for appropriately chosen $\beta_t$. This can be easily seen in the closed-form expression for Eq. (1): $\mathbf{x_t} = \sqrt{\bar{\alpha}_t}\mathbf{x_0} + \sqrt{1 - \bar{\alpha}_t}\boldsymbol{\epsilon}$. Here $\boldsymbol{\epsilon} \in \mathcal{N}(\boldsymbol{\epsilon}; \mathbf{0}, \mathbf{I})$, and we define $\alpha_t := 1 - \beta_t$ and $\bar{\alpha}_t := \prod_{s=1}^{t} \alpha_s$. DDPMs are therefore generative models that learn to recover the original data by modeling the reverse transitions:

$$p_\theta(\mathbf{x_{0:T}}) = p(\mathbf{x_T}) \prod_{t=1}^{T} p_\theta(\mathbf{x_{t-1}}|\mathbf{x_t}) , \quad p_\theta(\mathbf{x_{t-1}}|\mathbf{x_t}) = \mathcal{N}(\mathbf{x_{t-1}}; \mu_\theta(\mathbf{x_t}, t), \Sigma_\theta(\mathbf{x_t}, t)) . \quad (2)$$

Here $p_\theta(\mathbf{x_{t-1}}|\mathbf{x_t})$, parametrized by weights $\theta$, models a Gaussian distribution that reverses state $\mathbf{x_t}$ back to $\mathbf{x_{t-1}}$, and the base distribution is defined as the standard normal $p(\mathbf{x_T}) := \mathcal{N}(\mathbf{x_T}; \mathbf{0}, \mathbf{I})$. The variance $\Sigma_\theta$ is usually chosen to be $\sigma_t^2\mathbf{I}$, and can be learned or chosen empirically. Predictions from the DDPM can then be generated by iteratively applying the model and following the transitions $p_\theta(\mathbf{x_{t-1}}|\mathbf{x_t})$. We give more details on the sampling procedure in Section 2.3 and Appendix A.5.

The training objective can then be constructed in the context of variational inference, by approximating the posterior distribution of the latent variables $p_\theta(\mathbf{x_{1:T}}|\mathbf{x_0})$ by the forward process $q(\mathbf{x_{1:T}}|\mathbf{x_0})$. Using Bayes' rule to obtain $q(\mathbf{x_{t-1}}|\mathbf{x_t}, \mathbf{x_0})$, maximizing the evidence lower bound (ELBO) on $p_\theta(\mathbf{x_0})$ is then equivalent to minimizing the sum of T Kullback-Leibler (KL) divergences. It can be shown that this loss can then be rewritten as a simple minimization between true data and a denoising prediction (more details in Appendix A.1):

$$\mathcal{L}_{\text{DDPM}} = \mathbb{E}_{t, \mathbf{x_0}, \boldsymbol{\epsilon}} \left[ w_t ||\mathbf{x_0} - g_\theta(\mathbf{x_t}, t)||^2 \right] . \quad (3)$$

Here $g_\theta(\mathbf{x_t}, t)$ is a model that directly predicts $\mathbf{x_0}$ from $\mathbf{x_t}$, and $w_t$ is a weighting factor. Note that this integrates $t$ into the expectation; while the full loss should sum over all $t$, it is common

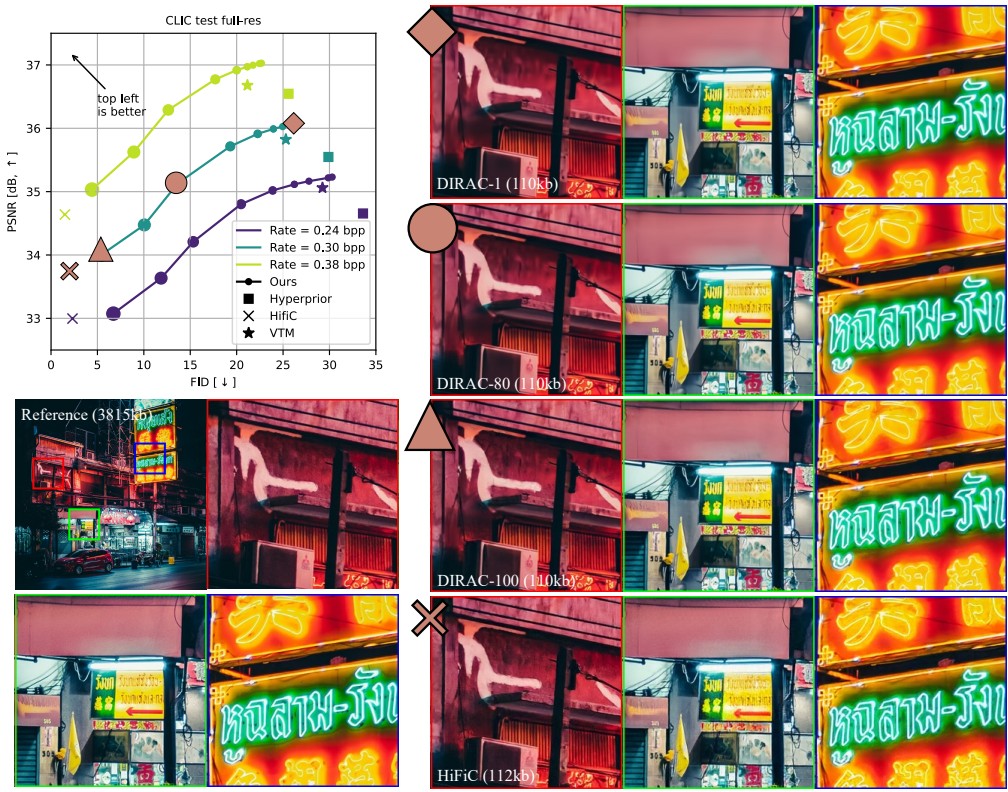

Figure 1: Top left: distortion-perception curves for DIRAC, a hyperprior baseline, a HiFiC baseline, and the standard codec VTM, showing that our codec is able to navigate a wide range of distortion-perception tradeoffs dynamically at test time (increasing marker size indicates number of diffusion steps). Bottom and right: Example reconstructions for various operating points, as well as the HiFiC baseline (cross). HiFiC is excellent at synthesizing textures, which often matches the contents of the image well (e.g. red/leftmost crop), but in some cases the textures appear unnatural (e.g. text in green/center crop). Image contents may be more perceptually pleasing when slightly blurrier (e.g. green neon text in blue/rightmost crop), so that a prediction with high fidelity (DIRAC-80, circle) also has higher perceptual quality. Our codec can adjust this tradeoff dynamically, e.g. to match user preferences. Rate matching of models is done using cubic spline interpolation.

practice to instead sample $t$ and perform Monte-Carlo integration over time. Ho et al. (2020) found that reweighting the individual loss terms can improve perceptual quality. The theoretically derived terms for $w_t$ become very for large small $t$ (see Appendix A.1 and Appendix A.3). We instead choose to set $w_t = 1$ to balance all terms evenly.

## 2.2 NEURAL DATA COMPRESSION

Neural network based codecs are systems that learn to compress data from examples. Most modern neural codecs are variations of *compressive autoencoders* (Theis et al., 2017), which transmit data using an autoencoder-like architecture consisting of three components. With a slight abuse of notation, let $\mathbf{x} = \mathbf{x_0}$ be the datapoint to compress. A neural encoder $E_\theta$ takes $\mathbf{x}$ and outputs a quantized latent variable $\mathbf{y}$. Given this latent variable, a neural decoder $D_\theta$ produces a reconstruction $\hat{\mathbf{x}}$. Typically, a neural prior $P_\theta$ models the distribution of latent variables $p(\mathbf{y})$, in order to losslessly compress latents using an entropy coding algorithm to $-\log p(\mathbf{y})$ bits in expectation. Neural codecs are typically trained using a rate-distortion objective consisting of two terms:

$$\mathcal{L}_{\text{RD}} = \mathbb{E}_{\mathbf{x} \sim p(\mathbf{x})} \left[ \mathcal{L}_{\text{distortion}}\big(\mathbf{x}, \hat{\mathbf{x}}(\mathbf{y})\big) + \lambda_{\text{rate}} \mathcal{L}_{\text{rate}}(\mathbf{y}) \right] . \tag{4}$$

The distortion loss $\mathcal{L}_{\text{distortion}}$ is a distance between the reconstruction $\hat{\mathbf{x}}$ and the ground truth $\mathbf{x}$, usually measured by mean squared error, $\mathcal{L}_{\text{distortion}}(\mathbf{x}, \hat{\mathbf{x}}) = |\mathbf{x} - \hat{\mathbf{x}}|^2$. The rate loss corresponds to the number of bits needed to transmit the quantized latent variable $\mathbf{y}$ under the neural prior, equal to $\mathcal{L}_{\text{rate}}(\mathbf{y}) = -\log P_\theta(\mathbf{y})$. The tradeoff parameter $\lambda_{\text{rate}}$ determines the expected compression ratio: for high $\lambda_{\text{rate}}$, few bits should be used, and vice versa. As it is impractical to train one codec per bitrate, a common choice is to condition the codec on the tradeoff parameter, and learn to operate at multiple bitrates by varying $\lambda_{\text{rate}}$ at training time (Song et al., 2021; Rippel et al., 2021).

## 2.3 A DIFFUSION-BASED NEURAL CODEC

**A DDPM-based decoder**  While the previous two sections were meant as an introduction to the necessary background, this section describes contributions made in this work. A DDPM can be constructed as a conditional model that exploits useful side-information, for example a target class or desirable latent features. In the data compression setting, one could build a DDPM conditioned on a compressed (quantized) latent representation $\mathbf{y}$ of the input $\mathbf{x}$, as is done in concurrent work (Yang & Mandt, 2022). In this work we choose to condition our diffusion model on an initial reconstruction $\tilde{\mathbf{x}}$ from a decoder $D_\theta^d$, which leads it to learn to improve perceptual quality, as we outline below.

Assuming a perfect encoder/decoder pair, the reconstruction $\tilde{\mathbf{x}}$ will have minimal distortion for the given bitrate. The DDPM $D_\theta^p$ has access to the same information as $D_\theta^d$, meaning that in expectation it cannot improve fidelity further. As the latent $\mathbf{y}$ is a compressed discrete representation, there are multiple images that result in the same latent, the preimage of $\mathbf{y}$: $p(\mathbf{x}|\mathbf{y}) = p(\mathbf{x}|\tilde{\mathbf{x}})$. An optimal DDPM will model the distribution $p(\mathbf{x}|\tilde{\mathbf{x}})$ exactly for any given $\tilde{\mathbf{x}}$. Although the perceptual metric that best corresponds to human perception is unknown, Blau & Michaeli (2019) formalize perceptual quality as a distance between the image distribution and the distribution of reconstructions. Perfectly modeling $p(\mathbf{x}|\tilde{\mathbf{x}})$ will trivially result in zero distance, and should thus lead to perfect perceptual quality under this definition. Note that this definition is also the basis of common perceptual metrics such as the Frechet Inception Distance (FID) (Heusel et al., 2017).

Furthermore, Blau & Michaeli (2019) show that rate, distortion and perception are in a triple trade-off. As our initial reconstruction $\tilde{\mathbf{x}}$ has the highest fidelity in expectation, and the bitrate is fixed, we expect an increase in perceptual quality to result in a decrease in fidelity. Intuitively this makes sense: we know that $\tilde{\mathbf{x}}$ minimizes the distortion to all $\mathbf{x} \sim p(\mathbf{x}|\tilde{\mathbf{x}})$, i.e., it is the distribution mean when distortion is measured in $L_2$. Samples from $p(\mathbf{x}|\tilde{\mathbf{x}})$ will have a non-zero expected distance from the mean, and thus reduced fidelity. As a result we have, in expectation, an initial reconstruction with optimal fidelity, and a DDPM-enhanced reconstruction with optimal perceptual quality.

Although in practice our encoder, decoder and DDPM are not optimal, we show that the distortion-perception tradeoff occurs empirically, and that intermediate samples from the DDPM will lead to a smooth traversal from a high fidelity image to one with high perceptual quality. We choose to only model residuals $\mathbf{r} = \mathbf{x} - \tilde{\mathbf{x}}$ with the diffusion model, which means the DDPM output is added to the initial reconstruction. This approach is visualized in Fig. 2. In theory this should be equivalent to predicting full images, but residuals follow an approximately Gaussian distribution, which we believe may be easier to model. More details on this choice are given in Appendix A.4.

**Optimization**  Our model is optimized using a rate-distortion-perception loss, a combination of the rate-distortion loss of Eq. (4) and the DDPM loss term of Eq. (3):

$$\mathcal{L}_{\text{RDP}} = \mathop{\mathbb{E}}_{\mathbf{x} \sim p(\mathbf{x})} \left[ \overbrace{\mathcal{L}_{\text{distortion}}(\mathbf{x}, \tilde{\mathbf{x}}) + \lambda_{\text{rate}} \mathcal{L}_{\text{rate}}(\mathbf{y})}^{\mathcal{L}_{\text{RD}}(\mathbf{x})} + \lambda_{\text{perception}} \mathcal{L}_{\text{DDPM}}(\mathbf{x}, \tilde{\mathbf{x}}) \right] \tag{5}$$

$$= \mathop{\mathbb{E}}_{\mathbf{x}, \boldsymbol{\epsilon}, t} \left[ ||\mathbf{x} - D_\theta^d(\mathbf{y})||^2 - \lambda_{\text{rate}} \log P_\theta(\mathbf{y}) + \lambda_{\text{perception}} ||\mathbf{r_0} - D_\theta^p(\mathbf{r_t}, t, \tilde{\mathbf{x}})||^2 \right] , \tag{6}$$

where we define the residual $\mathbf{r_0} = \mathbf{r} = \mathbf{x} - \tilde{\mathbf{x}}$, and the DDPM learns to model $p(\mathbf{r_0}|\mathbf{r_t}, \tilde{\mathbf{x}})$.

We find that training end-to-end with this objective does not work well. While end-to-end training of diffusion models with additional encoders has been demonstrated (Preechakul et al., 2022; Yang & Mandt, 2022), it is a common choice to perform training in separate stages (Rombach et al., 2022; Pandey et al., 2022). We train our codec in two stages, training only the base codec first using $\mathcal{L}_{\text{RD}}$,

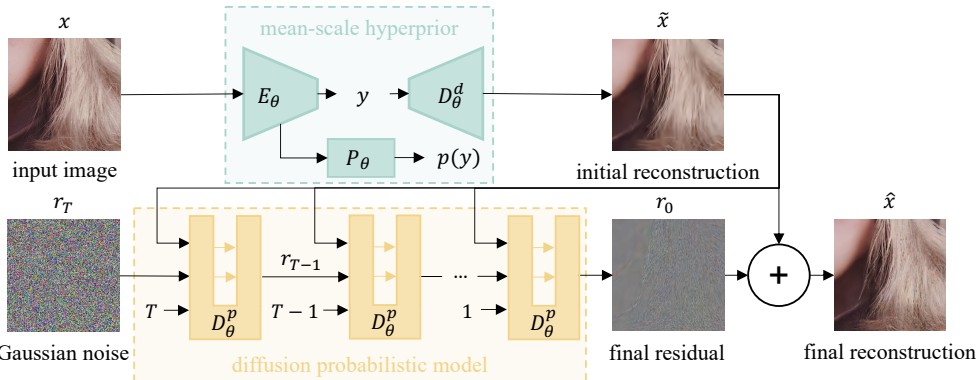

Figure 2: Overview of DIRAC, our proposed codec. A mean-scale hyperprior (optimized for distortion) transmits the input image $\mathbf{x}$ via a quantized latent $\mathbf{y}$, resulting in an initial reconstruction $\tilde{\mathbf{x}}$. This initial reconstruction is used as conditioning for a U-Net based DDPM (optimized for perceptual quality) to generate a residual $\mathbf{r_0}$. The final reconstruction is then $\hat{\mathbf{x}} = \tilde{\mathbf{x}} + \mathbf{r_0}$.

then freezing its parameters and training the DDPM using $\mathcal{L}_{\text{DDPM}}$. Final finetuning does not yield any improvements, so we omit it. Details on optimization parameters are given in Section 3.

**Fast sampling**  The computational cost of generation using diffusion models is often much higher than that of alternative models such as GANs (Ho et al., 2020), as $T$ model forward passes are required to sample one datapoint. Many efforts in this field were devoted to improving the sampling speed, see Section 5 for an overview. In this work, we use the deterministic sampling algorithm introduced by Denoising Diffusion Implicit Models (DDIM) (Song et al., 2020), as it is more robust to reducing the number of steps than DDPM sampling. We provide more context on this scheme in Appendix A.5. While we use $T = 1000$ during training, the number of sampling steps can be reduced to 100 at test time at negligible cost to performance, by redistributing the timesteps based on the scheme proposed by Nichol & Dhariwal (2021).

As mentioned earlier, the DDPM in DIRAC aims to improve perceptual quality at the cost of fidelity. We demonstrate in this work that the distortion-perception tradeoff can be navigated smoothly with the iterative DDIM sampling procedure: in practice, predicted residuals early in sampling have a low norm and are close to zero, and their norm increases gradually with each step. This means that the sampling procedure can be stopped at any point, e.g., when the desired perceptual quality is achieved, or when a compute budget is reached. To indicate how many sampling steps have been performed, we refer to our model as DIRAC-n, going from DIRAC-1 to DIRAC-100.

While early stopping navigates the perception-distortion tradeoff, we find that it is also possible to skip initial sampling steps, similar to observations by Dhariwal & Nichol (2021); Lyu et al. (2022). For the chosen noise schedule, early latents $\mathbf{x_t}$ are close to Gaussian noise. By inserting appropriately scaled noise as a "latent" at timestep $t$, we show in Section 4 that up to $80\%$ of the sampling procedure can be skipped with a negligible decrease in performance. The result is that our codec is vastly more practical than other diffusion-based approaches in the compression context. Additionally, it suggests that better noise schedules are possible in this setting.

## 3 EXPERIMENTS

**Baselines**  Our codec is able to dynamically adjust the perception-fidelity tradeoff at test time. We therefore compare it to the state-of-the-art methods in both rate-perception and rate-distortion performance. One of the strongest perceptual codecs is HiFiC (Mentzer et al., 2020). Our reimplementation obtains slightly better performance than their reported results. While neural codecs have obtained strong rate-distortion performance, the standard codec VTM (Bross et al., 2021) is still one of the best performing codecs in the low bitrate regime. We therefore include VTM as a rate-distortion baseline. Note that while our hyperprior base model (Ballé et al., 2018; Minnen et al., 2018) is not state-of-the-art, it is a well-established method for which high quality open source

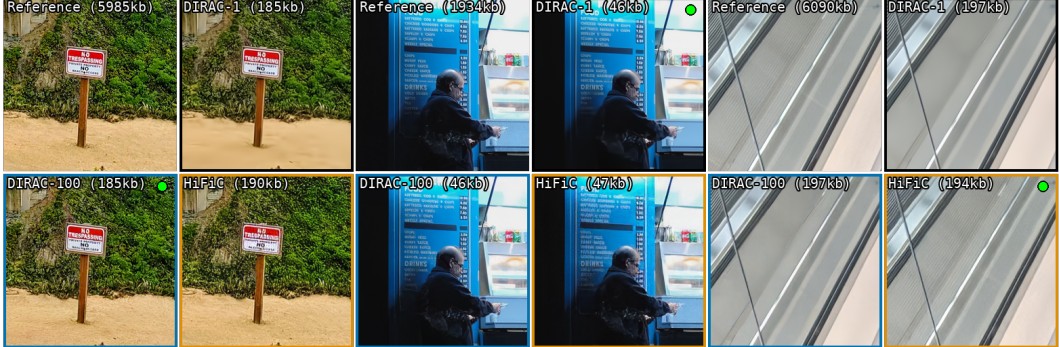

Figure 3: Selected CLIC2020 test set crops showcasing three cases where different distortion-perception operating points are preferred. Based on the authors' judgement, green dots indicate best subjective quality. Left shows how DIRAC-100 (percep. quality) fares better than HiFiC on text input, center shows how DIRAC-1 (fidelity) is beneficial for small text and faces, right shows HiFiC's superiority for fine texture generation. Best viewed electronically.

implementations are available. We also include the older JPEG codec (Wallace, 1991). Standard baselines are evaluated using the VTM reference software, CompressAI framework (Bégaint et al., 2020) and libjpeg in Pillow. We provide details on how to reproduce baselines in Appendix B.5.

**Datasets** For training, we use the training split of the high-resolution CLIC2020 dataset (Toderici et al., 2020) (1633 images of varying resolutions). We follow the three-step preprocessing pipeline of HiFiC (Mentzer et al., 2020): for each image, randomly resize according to a scale factor uniformly sampled from the range $[0.5, 1.0]$, then take a random $256 \times 256$ crop, then perform a horizontal flip with probability 0.5. For validation and model selection, we use the CLIC 2020 validation set (102 images). We test on two datasets: CLIC2020 test set (428 images) and the Kodak dataset (Kodak, 1991) (24 images), both common benchmarks for image compression. We evaluate on full resolution images: we reflect-pad the network input for sides to be multiple of 64—the total downsampling factor within the encoder and hyper-encoder—and crop the output back to the original resolution.

**Metrics** We evaluate our method using both distortion and objective perceptual quality measures. For distortion, we use the common PSNR and MS-SSIM (Wang et al., 2003) distortion measures. As perceptual quality measures, we use two different metrics: the full-reference LPIPS (Zhang et al., 2018) metric, and the Frechet Inception Distance (FID) (Heusel et al., 2017) metric, which measures the distance between the target distribution $p(\mathbf{x})$ and the distribution of reconstructions $p(\hat{\mathbf{x}})$. We also show Inception Score (Salimans et al., 2016), KID (Bińkowski et al., 2022) and NIQE (Mittal et al., 2013) in Appendix C.3. The latter requires a large dataset of images; following HiFiC evaluation, we take half-overlapping $256 \times 256$ patches from each image in a test dataset instead of resizing full-resolution images. All metrics are computed on images in the RGB color space. To report bitrates, we perform entropy coding and take the filesize. This leads to no more than $0.5\%$ overhead compared to the theoretical bitrate.

**Training details** We provide full details on the implementation in Appendix B, and report computational cost in Appendix B.4. We only repeat the most important details here. As mentioned earlier, we separate training into stages, which allows us to ignore the $\lambda_{\text{perception}}$ factor in Eq. (5): (1a) pre-train a single-rate mean-scale hyperprior (Minnen et al., 2018) for 2M iterations using the $\mathcal{L}_{\text{RD}}$ loss, (1b) finetune a multi-rate hyperprior for 1M iterations using the resulting weights, and (2) train the diffusion model for 650k steps using $\mathcal{L}_{\text{DDPM}}$, while keeping the hyperprior frozen. Training is performed using the Adam optimizer (Kingma & Ba, 2015) with a learning rate of $10^{-4}$ and no learning rate decay for the diffusion model, while learning rate is decayed by a factor of $0.5$ for the last 500k steps for the hyperprior model.

Our mean-scale hyperprior architecture follows (Ballé et al., 2018). It uses 4 downsampling/upsampling layers on the encoder/decoder, and 2 downsampling/upsampling layers on the

hyper-encoder/hyper-decoder. The model has 21.4 million parameters in total. For context, the HiFiC baseline has 181.5 million.

We base our implementation of DDPMs on the official open source implementation of Dhariwal & Nichol (2021)[1], and base most of our default architecture settings on the $256 \times 256$ DDPM from Preechakul et al. (2022), which uses a U-Net architecture (Ronneberger et al., 2015). This model has 108.4 million parameters. Conditioning on $\tilde{\mathbf{x}}$ is achieved by concatenating $\tilde{\mathbf{x}}$ and the DDPM latent $\mathbf{x_t}$. Early experiments revealed that certain architectural choices resulted in improved performance. To validate these, we perform an ablation study in Appendix B.1, using a smaller version of our model, and smaller crops of the validation set. In summary, we find that using the training objective of Eq. (3) (instead of the commonly used $\epsilon$ objective of Ho et al. (2020), see Appendix A.2), training on larger crops with increased model capacity results in large improvements in perceptual quality. Yet, the study also suggests our final model configuration may have room for further improvement.

## 4 RESULTS

**Distortion-perception tradeoff** In Fig. 1 we show the distortion-perception tradeoff for our model compared to the hyperprior base model and HiFiC. The sampling steps of the diffusion model gradually increase perceptual quality (lower FID), at the expense of fidelity. The figure visualizes this trajectory for an example image from the CLIC2020 test set. We find that a single sampling step from our model improves PSNR over the hyperprior base codec, likely due to larger model capacity and longer training time.

HiFiC performs best in FID, but we emphasize that depending on the setting, PSNR may be more important. Different content and applications require different distortion-perception operating points. To support this argument, we provide additional qualitative examples in Fig. 3. For each case, a different method is preferred between HiFiC, DIRAC-1 (highest fidelity) and DIRAC-100 (highest perceptual quality). While it is clear from the rightmost example that HiFiC is excellent at synthesizing texture, both left and middle example show that HiFiC hallucinations lead to poor results on information rich high frequency content like small text. DIRAC's dynamic test-time control of distortion-perception tradeoff allows mitigating the issue, as it spans close to the entire distortion-perception tradeoff, which makes it amenable to more use-cases. Additional qualitative results, as well as other approaches' reconstruction of the same examples, can be found in Appendix C.4.

**Comparison to state of the art** While the distortion-perception tradeoff is shown in Fig. 1, we visualize the individual rate-distortion and rate-perception tradeoffs in Fig. 4, showing both FID and LPIPS as measures of perceptual quality. Note that FID computation on Kodak is not reliable due to the low number of available samples, and we only show it for completeness. We show our model in two configurations: DIRAC-100 (100 sampling steps) has maximal perceptual quality, and DIRAC-1 (single sampling step) has minimal distortion. HiFiC demonstrates the best perceptual quality, but has poor performance in PSNR. In terms of PSNR, DIRAC-1 is close to VTM in the low-rate regime and beats it in the high-rate regime. We also show the concurrent work by Yang & Mandt (2022) for LPIPS on Kodak (scores extracted from their Fig. 2c, scores for other panels not available). Our model outperforms theirs by a large margin in the low-rate regime.

**Efficient sampling** In Fig. 5 we show the difference between sampling the full 100 steps (dotted line), and skipping the first 80 steps (solid line), on the CLIC2020 test set and for two different bitrates. The left and center panels show the change of PSNR and FID, respectively, as more sampling steps are performed. As seen in Fig. 1, both PSNR and FID decrease over time, traversing the distortion-perception tradeoff. Sampling can be stopped at any point to achieve the desired balance. For the lower rate example the change is more gradual, while the high-rate case shows little change during large parts of the sampling procedure. We further find that there is little difference in FID when we skip the first 80 sampling steps, while PSNR actually improves for the low-rate example. Note that the ability to start sampling from step 80 also means that all tradeoff points before that can be reached with a single one-shot prediction by our model, i.e., by scaling noise to the appropriate variance and performing a single diffusion step. These findings translate to the distortion-perception tradeoff in the rightmost panel, where we only show the low-rate example. DIRAC, using only the

---

[1]The official implementation of Dhariwal & Nichol (2021): https://github.com/openai/guided-diffusion

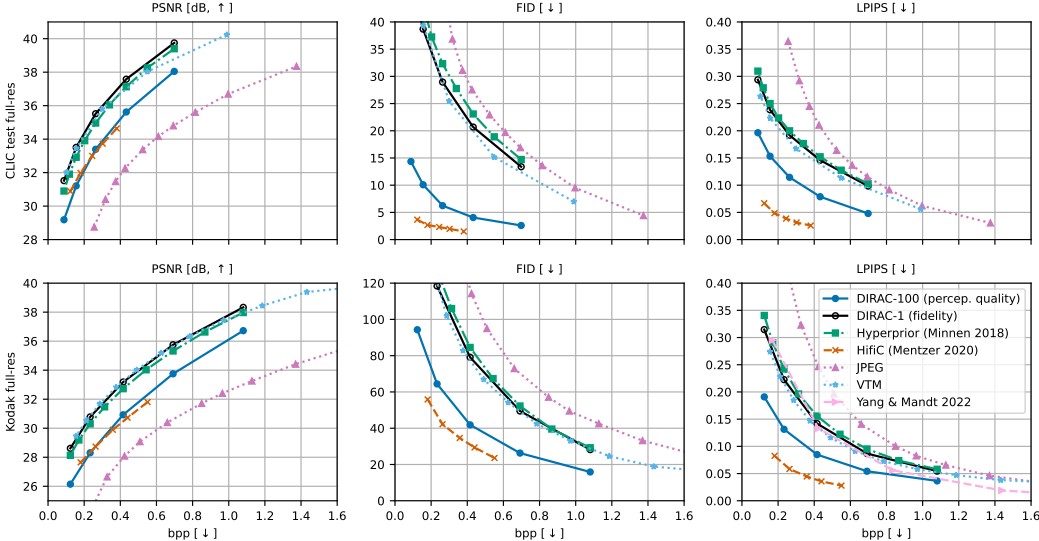

Figure 4: Rate-distortion (left) and rate-perception (middle & right) curves for the CLIC2020 test set (top) and Kodak dataset (bottom). Although FID on Kodak is based on a low number of crops, we report it here for completeness.

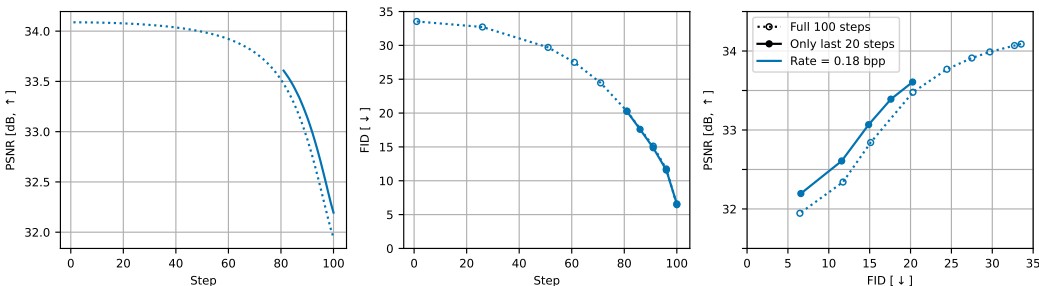

Figure 5: Analysis of the effect of early stopping and late starting on the CLIC2020 test set. We show tradeoffs for rate-distortion (left), rate-perception (center) and distortion-perception (right). The default sampling schedule of 100 steps is shown as a dotted line (invisible in case of perfect overlap), the solid line shows performance when skipping the first 80 steps.

last 20 sampling steps, actually performs better than the full sampling procedure in terms of PSNR, and the final FID is only negligibly worse. This behaviour is rate-dependent: at high bitrates results are virtually identical, at lower rates (shown in Fig. 5) we see an increase in PSNR.

## 5    RELATED WORK

**Neural data compression**    Neural codecs have seen major advances in both the image (Minnen et al., 2017; Rippel & Bourdev, 2017; Ballé et al., 2018; Minnen et al., 2018) and video domain (Wu et al., 2018; Lu et al., 2019; Rippel et al., 2019; Habibian et al., 2019; Agustsson et al., 2020; Hu et al., 2021; 2022). By conditioning on a $\lambda_{\text{rate}}$-variable corresponding to a desired bitrate, some codecs have been extended to operate at multiple bitrates (Wu et al., 2020; Song et al., 2021; Rippel et al., 2021). Perception-focused compression methods are mainly led by GAN-based approaches (Agustsson et al., 2019; Mentzer et al., 2020; Yang et al., 2021; Mentzer et al., 2021). Recent GAN-based works allow trading off fidelity and perceptual quality using control parameters and latent masking (Wu et al., 2020; Iwai et al., 2021). Unlike existing work that masks the latent to reduce the bitrate and determine which areas of the input are high fidelity, our codec does not entangle rate-distortion and perception-distortion tradeoffs.

**Diffusion probabilistic models** DDPMs have proved to be successful in various application areas, including image and video generation (Ho et al., 2022a;b; Yang et al., 2022), representation learning (Preechakul et al., 2022; Pandey et al., 2022), language-to-image generation (Ramesh et al., 2022; Saharia et al., 2022b), and image-to-image modelling such as image deblurring (Whang et al., 2022) and restoration (Kawar et al., 2022; Saharia et al., 2022a).

In the context of data compression, Hoogeboom et al. (2021) show that DDPMs can be used in the lossless compression setting. In the lossy compression setting, Ho et al. (2020) showed that if the continuous latents can be transmitted, a DDPM enables progressive coding. Theis et al. (2022) make this approach feasible in practice by using reverse channel coding. Although their approach is powerful and elegant, it is currently unlikely to be of practical use due to its high computational cost. In concurrent work, Yang & Mandt (2022) propose a codec where a conditional DDPM takes the role of the decoder, directly producing a reconstruction from a latent variable. In contrast, our codec first creates an initial reconstruction and then models the residuals between it and the ground truth, which enables direct control over perception-distortion and results in improved performance.

The sampling process for diffusion models is typically inefficient as it requires a high number of sampling steps to achieve the best performance. Although it is not our main focus, we show in this work that the compression setting lends itself well to speeding up sampling. Other works have shown that the diffusion process can be started at late timesteps (Lyu et al., 2022), and that this could mean that better-suited noise schedules exist (Nichol & Dhariwal, 2021). Various other methods were proposed to improve the sampling speed, either through model distillation (Luhman & Luhman, 2021; Salimans & Ho, 2021), by altering the sampling procedure (Song et al., 2020), or by working in a low-dimensional latent space (Rombach et al., 2022). In this work, we use DDIM (Song et al., 2020) as it provides a training-free sampling scheme based on a generalization of the forward process to non-Markovian diffusion processes. This choice of forward process leads to "shorter" Markov Chains, which increases sample efficiency.

## 6 DISCUSSION AND LIMITATIONS

Our generative codec navigates the rate-distortion-perception tradeoff dynamically at test time, meaning the user can determine the importance of bitrate, fidelity and perceptual quality themselves. Although this improves user control over the amount of hallucinated content, we currently do not control *where* such hallucinations occur. Similar to GAN-based codecs, we observe that increasing perception sometimes harms fidelity in small regions with semantically important content, such as faces and text. Addressing this limitation is an important next step for generative codecs.

Additionally, although we are able to drastically reduce the number of sampling steps, it is still fairly expensive to use a DDPM on the receiver side. The fact that we see an increase in PSNR after a single sampling step means that the base hyperprior codec is far from perfect, and that improving the base model or adopting a standard codec like VTM would increase the fidelity of the initial reconstruction. This likely means that the DDPM can be made smaller to reduce computational complexity. We also find that many sampling steps can be skipped without loss in performance. This suggests that improved noise schedules, or conditioning the DDPM on the noise so that different schedules can be used at test time (Whang et al., 2022), may further improve performance.

## 7 CONCLUSION

In this work, we propose a new neural image compression method called Diffusion-based Residual Augmentation Codec (DIRAC). This codec uses a variable bitrate base model to transmit an initial reconstruction with high fidelity to the original input, and then uses a diffusion probabilistic model to improve its perceptual quality. We show that this design choice enables fine control over the rate-distortion-perception tradeoff at test time, which for example enables users to choose if an image is decoded with high fidelity or high perceptual quality. Furthermore, although diffusion probabilistic models are notoriously expensive to sample from, we show that in the compression setting, the number of sampling steps can be reduced dramatically. Our work shows that diffusion-based codecs have compelling practical advantages that make them a promising candidate for high perceptual quality data compression.

**Ethics statement**   Neural codecs may be biased towards the data they have seen during training, for example leading to worse reconstructions for data with little support. Additionally, generative codecs may generate false but plausible details, which can be harmful for use cases where fidelity is important. We observe this especially when examining reconstructions containing text and faces. Deploying generative codecs will therefore require careful investigation of the desired operating points and their failure cases. By giving the user control over the distortion-perception tradeoff, our codec facilitates selection of this operating point at test-time.

**Reproducibility statement**   All datasets used in this work are publicly available, and we show our three-step data augmentation pipeline in Section 3. The proposed codec consists of two components: a base image codec, and DDPM that performs enhancement of the initial reconstruction.

The base image codec is a mean-scale hyperprior (Ballé et al., 2018). High quality implementations of this neural codec (and similar image codecs) are available via CompressAI, see for example this mean-scale hyperprior implementation link on GitHub. This library also provides entropy coding functionality. Relevant hyperparameters for hyperprior training are listed in Appendix B.1.

The DDPM component was trained using the open source implementation of (Dhariwal & Nichol, 2021). The main change in implementation is that our DDPM is conditioned on an initial reconstruction from a mean-scale hyperprior, which is achieved by concatenating it with the DDPM latent (these two tensors have the same spatial dimensions). We list all hyperparameter settings in Table 1, as well as the name of command line arguments corresponding to those parameters. We provide details on sampling in Appendix A.5, Appendix B.3.

Lastly, we provide information about computational complexity and training compute in Appendix B.4. Instructions for reproducing baselines are provided in Appendix B.5.

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

# A  METHOD

## A.1  DERIVATION OF DDPM LOSS FOR LEARNING $\mathbf{x_0}$

Following Sohl-Dickstein et al. (2015), the ELBO on the log likelihood of $p_\theta(\mathbf{x_0})$ can be written as $T$-step Kullback–Leibler (KL) divergences: $KL(q(\mathbf{x_{t-1}}|\mathbf{x_t},\mathbf{x_0})||p_\theta(\mathbf{x_{t-1}}|\mathbf{x_t}))$ for $t = 1,...,T-1$. Under the assumption that the two distributions of interest are both Gaussian, we further assume $\Sigma_\theta(\mathbf{x_t},t) = \sigma_t^2\mathbf{I}$ to be time dependent constants. This reduces the KL divergence at step $t$ to a comparison from the model mean to the posterior mean of the forward process, i,e.,

$$KL(q(\mathbf{x_{t-1}}|\mathbf{x_t},\mathbf{x_0})||p_\theta(\mathbf{x_{t-1}}|\mathbf{x_t})) = \mathbb{E}_q\left[\frac{1}{2\sigma_t^2}||\tilde{\boldsymbol{\mu}}_t(\mathbf{x_t},\mathbf{x_0}) - \boldsymbol{\mu}_\theta(\mathbf{x_t},t)||^2\right] + C, \tag{7}$$

where $C$ is a constant, and $\tilde{\boldsymbol{\mu}}_t$ denotes the posterior mean of the forward process conditioned on the data input $\mathbf{x_0}$, i.e., $q(\mathbf{x_{t-1}}|\mathbf{x_t},\mathbf{x_0})$.

Note that, using the Bayes' rule and the Markovian property of the forward process, we can rewrite

$$q(\mathbf{x_{t-1}}|\mathbf{x_t},\mathbf{x_0}) = \frac{q(\mathbf{x_{t-1}},\mathbf{x_t}|\mathbf{x_0})}{q(\mathbf{x_t}|\mathbf{x_0})}$$
$$= \frac{q(\mathbf{x_{t-1}}|\mathbf{x_0})q(\mathbf{x_t}|\mathbf{x_{t-1}})}{q(\mathbf{x_t}|\mathbf{x_0})}, \tag{8}$$

where the distribution $q(\mathbf{x_t}|\mathbf{x_0}) = \mathcal{N}(\mathbf{x_t}; \sqrt{\bar{\alpha}_t}\mathbf{x_0}, (1-\bar{\alpha}_t)\mathbf{I})$ with $\alpha_t := 1-\beta_t$ and $\bar{\alpha}_t := \prod_{s=1}^{t}\alpha_s$. With the known distribution of $q(\mathbf{x_t}|\mathbf{x_0})$, we can sample $\mathbf{x_t}$ at an arbitrary step $t$ by

$$\mathbf{x_t}(\mathbf{x_0},\boldsymbol{\epsilon}) = \sqrt{\bar{\alpha}_t}\mathbf{x_0} + \sqrt{1-\bar{\alpha}_t}\boldsymbol{\epsilon}, \quad \text{where} \quad \boldsymbol{\epsilon} \sim \mathcal{N}(\mathbf{0},\mathbf{I}). \tag{9}$$

Now each of the components on the RHS of Eq. (8) is defined as a Gaussian distribution with known parameters. As a result, we can find the posterior mean $\tilde{\boldsymbol{\mu}}_t$ in the following explicit form:

$$\tilde{\boldsymbol{\mu}}_t(\mathbf{x_t},\mathbf{x_0}) := \frac{\sqrt{\bar{\alpha}_{t-1}}\beta_t}{1-\bar{\alpha}_t}\mathbf{x_0} + \frac{\sqrt{\alpha_t}(1-\bar{\alpha}_{t-1})}{1-\bar{\alpha}_t}\mathbf{x_t} = \eta_t\mathbf{x_0} + \xi_t\mathbf{x_t}. \tag{10}$$

By plugging Eq. (10) into Eq. (7), and choosing a parametrization that matches $\boldsymbol{\mu}_\theta$ to $\tilde{\boldsymbol{\mu}}_t$ at each diffusion step $t$, for example $\boldsymbol{\mu}_\theta(\mathbf{x_t},t) = \eta_t g_\theta(\mathbf{x_t},t) + \xi_t\boldsymbol{\epsilon}$ (where $g_\theta$ is a function that directly predicts $\mathbf{x_0}$ from $\mathbf{x_t}$), we arrive at an objective function at diffusion step $t$:

$$\min_{t,\mathbf{x_0},\boldsymbol{\epsilon}} \mathbb{E}\left[w_t||\mathbf{x_0} - g_\theta(\mathbf{x_t},t)||^2\right] \quad \text{for} \quad t = 1,...,T-1. \tag{11}$$

where $w_t := \eta_t^2/2\sigma_t^2$ and $t$ is absorbed into the expectation instead of being summed over.

## A.2  PARAMETRIZATION FOR LEARNING THE NOISE $\epsilon$

For completeness, we use this section to restate the approach of Ho et al. (2020), who parametrize models to predict the noise $\boldsymbol{\epsilon}$ instead of the data $\mathbf{x_0}$.

Ho et al. (2020) reparameterized Eq. (9) to obtain a function of $\mathbf{x_0}$: $\mathbf{x_0}(\mathbf{x_t},\boldsymbol{\epsilon}) = \frac{1}{\sqrt{\bar{\alpha}_t}}(\mathbf{x_t} + \sqrt{1-\bar{\alpha}_t}\boldsymbol{\epsilon})$ where $\boldsymbol{\epsilon} \sim \mathcal{N}(\mathbf{0},\mathbf{I})$. To minimize the KL divergence (7) at time $t$, the authors chose a formulation that learns the noise $\boldsymbol{\epsilon}$, which leads to the following desired equality:

$$\boldsymbol{\mu}_\theta(\mathbf{x_t},t) = \tilde{\boldsymbol{\mu}}_t(\mathbf{x_t}, \frac{1}{\sqrt{\bar{\alpha}_t}}(\mathbf{x_t} + \sqrt{1-\bar{\alpha}_t}\boldsymbol{\epsilon}_\theta)). \tag{12}$$

Expanding the posterior mean according to Eq. (10), we can derive an objective function that minimizes prediction errors in the $\boldsymbol{\epsilon}$-space:

$$\mathcal{L}_\epsilon = \mathbb{E}_{\mathbf{x_0},\boldsymbol{\epsilon}}\left[v_t||\boldsymbol{\epsilon} - \boldsymbol{\epsilon}_\theta(\sqrt{\bar{\alpha}_t}\mathbf{x_0} + \sqrt{1-\bar{\alpha}_t}\boldsymbol{\epsilon},t)||^2\right], \tag{13}$$

where the weight coefficients are $v_t := \beta_t^2/(2\sigma_t^2\alpha_t(1-\bar{\alpha}_t))$.

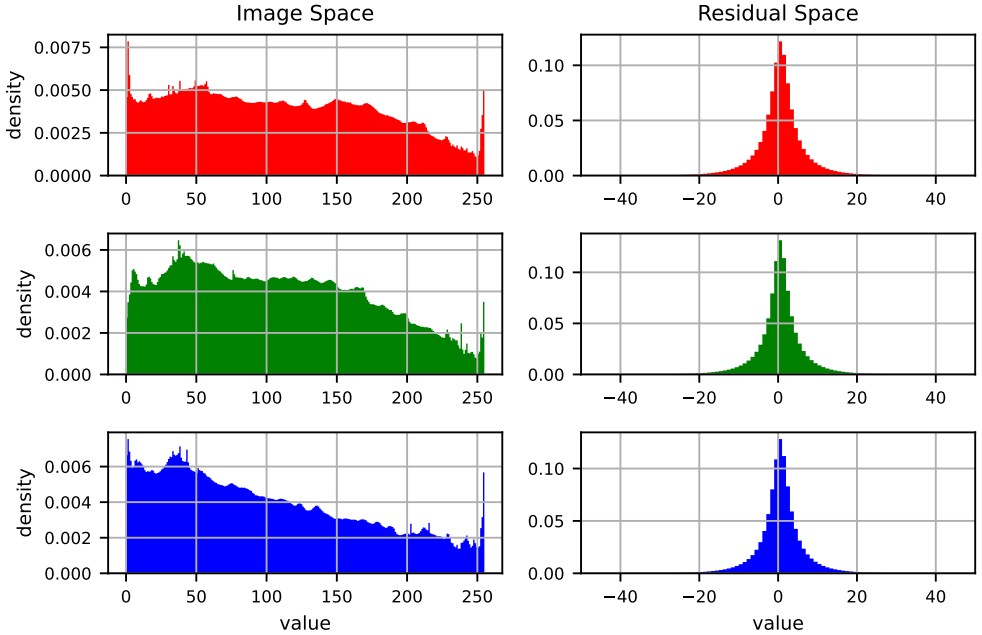

Figure 6: Distributions of pixel values in a 1000 random 256x256 crops of the CLIC train dataset, left for the target image $\mathbf{x}$, right for the residual $\mathbf{r} = \mathbf{x} - \tilde{\mathbf{x}}$ where the initial reconstruction $\tilde{\mathbf{x}}$ is from a single-rate mean-scale hyperprior trained with $\lambda_{\text{rate}} = 0.03$. Rows corresponds to the red, green and blue channels respectively.

### A.3 NOISE SCHEDULE AND LOSS WEIGHTS

We choose the noise schedule $\beta_t$ such that the $\bar{\alpha}_t$ follow a cosine-shaped curve. This is one of the most common choices for the noise schedule. We also reweight the individual loss terms from Eq. (3) to $w_t = 1$, inspired by Ho et al. (2020) who do the same for loss formulation given in Eq. (13) (meaning they set $v_t = 1$). The theoretically derived $w_t = \eta^2/(2\sigma_t^2)$ results in extremely large weights for small $t$. We find that $w_t = 1$ works better, likely because of the more evenly distributed loss scales, but it is possible that better weights exist.

### A.4 MOTIVATION FOR DDPM MODELING IN RESIDUAL SPACE

DIRAC outputs an initial reconstruction $\tilde{\mathbf{x}}$, then models the distribution of residuals $p(\mathbf{r}|\tilde{\mathbf{x}})$ using a DDPM, where $\mathbf{r} = \mathbf{x} - \tilde{\mathbf{x}}$. In theory, one could model the image distribution $p(\mathbf{x}|\tilde{\mathbf{x}})$ directly. From an entropy perspective, the two approaches are equal: the entropy of $p(\mathbf{r}|\tilde{\mathbf{x}})$ is equal to the entropy of $p(\mathbf{x}|\tilde{\mathbf{x}})$ since, given the data $\mathbf{x}$, the residual $\mathbf{r}$ is a deterministic function of the initial reconstruction. Yet, while their entropy might be similar, the shape of these distributions is different.

In Fig. 6, we show the histogram of pixel values in the target image (left) and the residuals (right) for each RGB channel. We observe that residual values approximately follow a normal distribution, whereas the pixel values in image space show a more intricate distribution. Given the fact that DDPMs (in their typical formulation) map Gaussian noise to the target distribution, we conjecture that it is desirable that the target distribution is close to a normal distribution, and that it helps reduce the number of sampling steps required to obtain satisfactory perceptual quality.

### A.5 SAMPLING DESCRIPTION

We use 100 DDIM sampling steps by default (Song et al., 2020). We describe modified sampling schemes, namely the early stopping and late start procedures, in more detail. Both schemes speed up sampling by skipping steps entirely.

**Early stopping** The DDPM enhancement model of DIRAC predicts the (often sparse) residual $\mathbf{r} = \mathbf{x} - \tilde{\mathbf{x}}$. We find that we are able to stop sampling at any point before we reach the final step, by simply using the intermediate prediction for the residual $\hat{\mathbf{r}}_0(\mathbf{r}_t) = D_\theta^p(\mathbf{r_t}, t, \tilde{\mathbf{x}})$ as the final sample. Specifically, we observed that during sampling, early intermediate predictions are close to zero, and that they become sharper over time. Stopping early then typically means higher PSNR, whereas stopping late results in high perceptual quality. Intuitively, this makes sense as well: we know that in the limit of perfect models, $\tilde{\mathbf{x}}$ has the highest possible expected fidelity, and the DDPM can only increase quality by decreasing fidelity.

Stopping early in this manner is somewhat similar to the scheme proposed by Ho et al. (2020), who use this intermediate prediction to describe a progressive coding scheme. However, we have already transmitted the initial reconstruction $\tilde{\mathbf{x}}$, and all DDPM sampling happens on the receiver side, so the settings are quite different. It is the receiver-side generation of residuals that allows to navigate the the distortion-perception tradeoff (Blau & Michaeli, 2019), and that enables early stopping to achieve a desired tradeoff or to reduce compute requirements. Optimizing sampling trajectories of diffusion models is an active field, and further research on the sampling in different (conditional) settings may lead to actionable insights (Deja et al., 2022).

**Late start** Similar to findings of Nichol & Dhariwal (2021) and Lyu et al. (2022), we observe that it is possible to skip several sampling steps. In particular, we take an initial sample $\epsilon \sim \mathcal{N}(0, 1)$ and scale it to match the expected standard deviation at timestep $t$ given by the forward process, then plug this "latent" into the reverse process. Nichol & Dhariwal (2021) mainly use the late-start observation to motivate the use of a different noise schedule. Based on the similarity of our observations to theirs, we believe there may be noise schedules that are better suited for the image compression setting, but we leave that investigation for future work. The key finding in our work is that we can skip a large part (up to 80%) of the initial steps without performance degradation.

# B IMPLEMENTATION DETAILS

## B.1 ARCHITECTURE AND TRAINING

**Base codec** We use the "mean-scale hyperprior" architecture from Minnen et al. (2018). This is an hierarchical VAE with quantized latent variables. The first level encoder and decoder produce the quantized latent variable and decode it to a reconstruction. The second level, which is the prior model, uses a hyper-encoder to produce a so-called quantized hyper-latent $\mathbf{z}$, then maps that to parameters $\mu_{\mathbf{y}}, \sigma_{\mathbf{y}}$ using a hyper-decoder. The hyper-latent distribution is typically modeled using an unconditional prior $p(\mathbf{z})$. The probability of the quantized latent under the prior is then equal to $p(\mathbf{y}|\mathbf{z}) = \mathcal{N}(\mathbf{y}|\mu_{\mathbf{y}}, \sigma_{\mathbf{y}})$.

The encoder consist of 4 stride 2 convolutions with kernel size 5, and similar transpose convolution layers for the decoder. The hyper-encoder consists of a stride 1 convolution with kernel size 3, followed by 2 stride 2 convolutions with kernel size 5. We use 320 channels for every layer in the encoder/decoder, and 192 channels for the hyper-encoder/hyper-decoder.

As the codec uses $4 + 2$ stride 2 convolutions, it has a so-called downsampling factor of $2^{4+2} = 64\times$. To enable transmission of images that have spatial dimensions not divisible by this factor, we use 'reflect' padding to pad the image, transmit the padded image, then crop to the original resolution on the receiver side. We transmit the original spatial dimensions as 16 bit integers. This bit cost is negligible compared to the cost of transmitting the content.

Latent quantization is performed using a "mixed" strategy: both decoders see the quantized latents during training, and a straight-through estimator is used to make sure gradients pass through the hard quantization operation; the prior computes the loss based on latents quantized using additive uniform quantization noise $\mathbf{u} \sim U(-0.5, 0.5)$ (Guo et al., 2021). For more details and architecture visualizations, we refer the reader to Ballé et al. (2018); Minnen et al. (2018).

**Multi-rate training** Multi-rate capabilities are added using a scheme similar to that of Song et al. (2021); Rippel et al. (2021), by conditioning on $\lambda_{\text{rate}}$ during training and evaluation.

During training, the multi-rate codec must see range of tradeoff parameters in order to learn the desired effect of $\lambda_{\textbf{rate}}$. For each batch, we sample a value $\lambda' \sim U[0, 1]$, then skew this by a factor $\gamma$ to obtain $\lambda'' = \lambda'^\gamma$. We set $\gamma = 3.0$ to favor sampling of high bitrate batches, as we found that this improves fidelity for low bitrate settings as well. The final sampled tradeoff parameter $\lambda_{\text{rate}}$ is then obtained via linear interpolation in log space:

$$\log_2 \lambda_{\text{rate}} = (\log_2(\lambda_{\max}) - \log_2(\lambda_{\min})) \times \lambda'' + \log_2(\lambda_{\min}) , \tag{14}$$

where $\lambda_{\max} = 0.0128$ and $\lambda_{\min} = 0.0001$. $\lambda_{\text{rate}}$ is used for the loss, while $\lambda''$ is provided to the base codec. It is embedded in a soft one-hot vector of 4 dimensions $\vec{\lambda}$, following (Rippel et al., 2021). For example, this means the vector $[1, 0, 0, 0]$ corresponds to the value $\lambda'' = 1$, and $[0, 0.5, 0.5, 0]$ corresponds to $\lambda'' = 0.5$.

Each layer in the codec (sender and receiver side) is conditioned on the tradeoff parameter using output activation scale-and-shift, following (Song et al., 2021). This works as follows: for each activation in the codec $h_{\text{act}}$, which has $C_{\text{act}}$ channels, a linear layer projects the 4-element tensor $\vec{\lambda}$ onto a tensor with $2 \cdot C_{\text{act}}$ elements. The first $C_{\text{act}}$ elements correspond to the scale $s_{\text{act}}$, the remainder corresponds to the shift $d_{\text{act}}$ The conditioned output activation is then equal to $h_{\text{act}} \cdot s_{\text{act}} + d_{\text{act}}$.

At test time, the user picks the rate-distortion operating point by selecting a $\lambda_{\text{rate}}$ value. Conditioning is then performed in the same way as during training. Of course, we need to transmit $\lambda_{\text{rate}}$ to the receiver end as well. To do so, we quantize $\lambda''$ to a 16 bit integer, and transmit it at negligible bitrate cost.

**DDPM architecture**   Many of our U-Net architecture choices were adopted from the open-source implementation of Dhariwal & Nichol (2021), and we refer the reader to those works for a detailed explanation of the meaning of all hyperparameters. Table 1 provides an overview of the hyperparameter settings for the full DIRAC model, as well as the DIRAC-small, which is the base model for our ablation study. Horizontal lines separate U-Net parameters, diffusion process parameters, and training hyperparameters.

The channel multiplier corresponds to the increase in width with respect to the base number of channels in each layer. Following Preechakul et al. (2022), we train our largest models using a U-Net with 6 separate blocks, increasing the multiplier every 2 blocks. Self-attention layers are removed from all layers except the bottleneck of the U-Net. We use the cosine noise schedule of (Nichol & Dhariwal, 2021), as we saw better performance with this schedule in early experiments. However, an ablation study in Appendix B.2 shows that even better hyperparameters may exist.

**Training DIRAC**   We separate training into two steps: 1) train our base codec for a rate distortion objective Eq. (4), and 2) then train the DDPM using the perceptual objective of Eq. (3).

Step 1 is split into two steps as well. First, we train a single rate hyperprior for 2 million iterations using $\lambda_{\text{rate}} = 0.001$. Then, we add $\lambda_{\text{rate}}$ conditioning layers, and finetune the multi-rate hyperprior for 1M additional iterations, as described in the paragraph "Multi-rate training". Although rate-distortion performance is important, the choice of base codec likely is not critical, and any multi-rate image codec may suffice.

In step 2, we use the base codec to produce initial reconstructions $\tilde{x}$, and train a $\tilde{x}$-conditional DDPM to perform enhancement. In order to support the multi-rate hyperprior, the DDPM needs to see reconstructions for many different $\lambda_{\text{rate}}$ at training time. In practice, this can be achieved by using the sampling schedule used for multi-rate hyperprior training. The DDPM is trained for 650,000 iterations, see the hyperparameter settings specified in Table 1.

## B.2   ARCHITECTURE ABLATION STUDY

We adopt most architecture settings for the DDPM from Dhariwal & Nichol (2021), but observed that certain architectural choices provided a significant performance improvement in early experiments on low-resolution data. To validate this, we ablate some architecture choices in Figure 7. Ablation studies were performed on a smaller reference model called DIRAC-small (details in Table 1) to enable training on a single V100 GPU, whereas DIRAC is trained on 2 A100 GPUs. For

| Parameter | Command line | DIRAC | DIRAC-small |
|---|---|---:|---:|
| Channel multiplier | `channel_mult` | 1,1,2,2,4,4 | 1,2,3 |
| Base num. channels | `num_channels` | 128 | 32 |
| Learn $\sigma_t$ | `learn_sigma` | false | false |
| Group normalization | `group_norm` | true | true |
| Attention resolutions | `attention_resolutions` | none | none |
| Num. attention heads | `num_heads` | 1 | 1 |
| Objective Eq. (3) | `predict_xstart` | true | true |
| Num. ResBlocks | `num_res_blocks` | 2 | 2 |
| Scale shift conditioning | `use_scale_shift_norm` | false | false |
| Diffusion steps | `diffusion_steps` | 1000 | 1000 |
| Noise schedule | `noise_schedule` | cosine | cosine |
| Batch size | `batch_size` | 128 | 32 |
| Learning rate | `lr` | $10^{-4}$ | $10^{-4}$ |
| Training steps | `iterations` | 650k | 600k |
| Input resolution | `image_size` | 256x256 | 256x256 |
| Use DDIM sampling | `use_ddim` | true | true |
| Resampling timesteps | `timestep_respacing` | ddim100 | ddim100 |

Table 1: Hyperparameters for DIRAC and the smaller ablation model during training and sampling. We refer the reader to the official open source implementation of Dhariwal & Nichol (2021) for more details on these parameters.

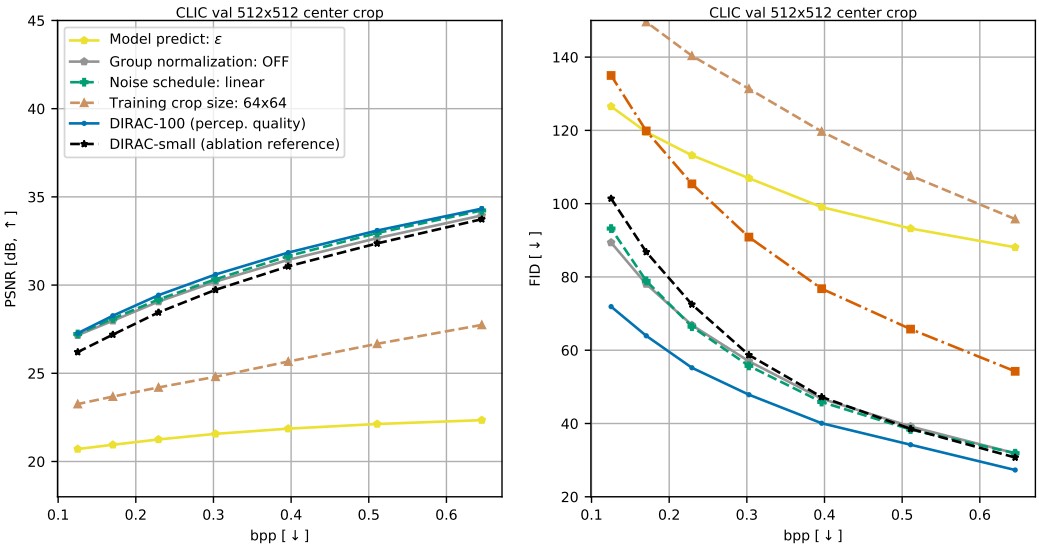

Figure 7: Architecture ablation study for DDPM model. Ablation reference (DIRAC-small) model is defined in Table 1.

all ablations, we sample using DDIM, reducing sampling to 100 timesteps, in a similar manner to Nichol & Dhariwal (2021). We evaluate PSNR and FID metrics on 512x512 center crops of the CLIC validation dataset.

We make the following observations. Training on smaller crops, using the $\epsilon$ objective of Ho et al. (2020) instead of the $\mathbf{r_0}$ objective of Eq. (3) results in much worse performance. However, disabling group normalization and using a linear noise schedule both improve performance. Our final model does not use these settings, meaning further improvements may be possible for DIRAC. The full DIRAC model performs better than DIRAC-small, especially in FID at lower bitrates, meaning that

increased model capacity likely contributes to performance. Nevertheless, using a smaller DDPM is possible if compute cost is more important than FID.

### B.3 SAMPLING

We use DDIM sampling with 100 timesteps for all experiments, following Song et al. (2020), unless stated otherwise. Sampling for 100 timesteps instead of the 1000 used during training requires "timestep respacing" of the noise schedule, which we achieve using the scheme in proposed in Nichol & Dhariwal (2021).

To enable sampling for images with shapes not divisible by the downsampling factor of the U-Net, we pad the conditioning using 'reflect' padding, sample an image of the resulting size, then crop to the original spatial dimensions. This does not affect the bitrate, as all information already lives on the receiver end.

As mentioned earlier, for all early stopping and late start experiments, we use the intermediate timestep $t$ prediction $\hat{\mathbf{r}}_0(\mathbf{r}_t) = D_\theta^p(\mathbf{r_t}, t, \tilde{\mathbf{x}})$ as the best-so-far prediction. This is the only sensible choice: the latent $\mathbf{r}_t$ is still close to random noise at earlier iterations.

### B.4 COMPUTE

We provide information on the computational complexity of model components, and the cost of sampling, in Table 2. We run benchmarking on a desktop workstation with a Nvidia 3080 Ti card, CUDA driver version is 455.32.00, CUDA 11.1. We use 1,000 forward passes on square inputs of size $256 \times 256$ and $1024 \times 1024$, use GPU warmup, and record inference time using CUDA events. We do not include time to entropy code here.

|  | Parameter count | Runtime 256 (ms) | Runtime 1024 (ms) |
|---|---|---|---|
| DIRAC hyperprior | 21.4M | $13.9 \pm 1.0$ | $82.7 \pm 1.81$ |
| DIRAC U-Net | 108.4M | $39.3 \pm 1.2$ | $485.6 \pm 1.17$ |
| HiFiC | 181.5M | $18.4 \pm 1.1$ | $129.2 \pm 0.64$ |

Table 2: Computational complexity for each model. Runtime mean and standard deviation were obtained using 1,000 forward passes. Note that none of the models here were explicitly optimized for inference speed, numbers are indications only.

All our (base codec) multi-rate hyperpriors were trained on a single Nvidia V100 card. Final DIRAC models were trained on 2 Nvidia A100 cards using data paralellism. Ablation model (DIRAC-small) settings were chosen so that training could be performed on one Nvidia V100 card.

A fair comparison between methods would not only require equalization of bitrate, but ideally also equalization of training and test-time compute. For example, HiFiC decoding complexity is substantially lower than that of DIRAC if many sampling steps are used, which will give DIRAC-100 an unfair advantage. DDPMs also see more datapoints than HiFiC during training as a larger batch size is used. In this work, we mainly focused on feasibility of the approach, and have therefore not equalized compute.

### B.5 BASELINES

**Standard codec baselines**  We run VTM-17.0 `https://vcgit.hhi.fraunhofer.de/jvet/VVCSoftware_VTM/-/releases/VTM-17.0`, and use CompressAI `https://github.com/InterDigitalInc/CompressAI` to prepare the encoding command. CompressAI converts given input RGB images to YUV444 before coding them in "all intra" mode, then converts the reconstructed YUV444 images back to RGB. These conversions are lossless. We use the default all intra configuration, and use QPs $\{22, 27, 32, 37, 40\}$, where higher QPs corresponds to low bitrate and vice versa.

Specifically, the command used to gather VTM-17.0 evaluations is:

```
python -m compressai.utils.bench vtm <path-to-image-folder>
-c <path-to-VTM17.0-software>/cfg/encoder_intra_vtm.cfg
-b <path-to-VTM17.0-software>/bin
-q [22, 27, 32, 37, 40]
```

For JPEG evaluations, we use JPEG functionality in Pillow. We again use CompressAI to prepare the encoding command.

```
python3 -m compressai.utils.bench jpeg <path-to-image-folder>
-q [5, 10, 15, 20, 25, 30, 35, 40, 45, 55]
```

**Hyperprior baselines**  We train mean-scale hyperpriors for various rate-distortion tradeoff points by setting $\lambda_{\text{rate}}$ to one of $\{0.006, 0.005, 0.004, 0.003, 0.002, 0.001, 0.0003\}$, where 0.006 corresponds to the lowest bitrate model. Baseline models were trained for up to 2 million iterations, where we select the best model based on Kodak PSNR. Selecting baseline models based on their test dataset performance may be unfair to other methods, but ensures the baseline is strong.

**HiFiC baselines**  We train HiFiC models, as described by (Mentzer et al., 2020), for 5 bitrate targets: $\{0.15, 0.225, 0.3, 0.375, 0.45\}$. Each model is trained in two stages. In the first stage, we train a mean-scale hyperprior with a rate-distortion objective for 1 million iterations, where the distortion objective combines mean squared error with the perceptual distortion metric LPIPS (Zhang et al., 2018). In the second stage, a conditional discriminator is added and the model is finetuned for an additional 1 million iterations using a rate-distortion-GAN loss. In both stages, learning rate is decayed from $10^{-4}$ to $10^{-5}$ after 500,000 iterations. We use the models at the final timestep (2 million iterations) as baseline models.

We follow the training procedure described in the paper, with two important deviations. First, we use $\lambda_{\text{rate}}$ warmup for 100K iterations, instead of the 50K iterations used in the paper. Second, we enable this warmup schedule for GAN finetuning as well. We find that these changes result in slightly improved performance with respect to the scores reported by Mentzer et al. (2020), especially at low bitrate.

## C  ADDITIONAL RESULTS

### C.1  ROBUSTNESS TO DIFFERENT BASE CODECS

In Fig. 8 we apply DIRAC-100 (maximum perceptual quality), which was trained with a hyperprior base codec, directly on top of other base codecs, without finetuning. Specifically, we use JPEG and VTM as examples of standard codecs, and SwinT (Zhu et al., 2022) as a state-of-the-art neural codec. We find that our model still improves perceptual quality, indicating that it is robust to the choice of base codec. It is reasonable to assume that training on these base codecs would improve performance further. On VTM and SwinT, the gain in perceptual quality is still large, close to what we achieve for the hyperprior base codec. Using JPEG, the increase is only moderate. We believe this is due to the fact that JPEG exhibits strong artifacts at low rates, specifically blocking. Reconstructions from VTM and SwinT, on the other hand, are pretty similar to those from the hyperprior, in that low-rate reconstructions tend to be blurry.

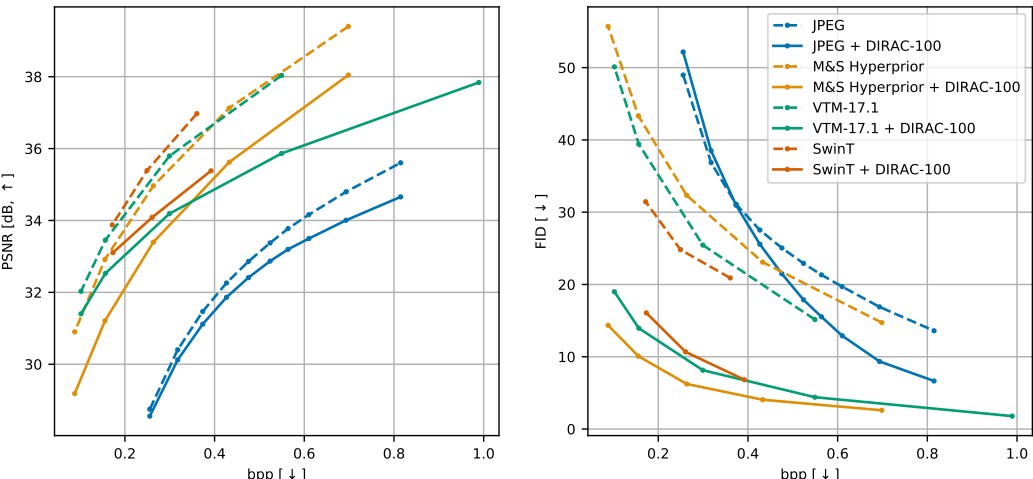

Figure 8: We apply DIRAC-100, trained with a mean-and-scale hyperprior base codec, to reconstructions from other base codecs, namely JPEG and VTM-17.1, on the CLIC2020 test set. Without finetuning, our model can still tradeoff fidelity and perceptual quality effectively.

## C.2 COMPARISON TO IMAGE ENHANCEMENT APPROACHES

As we do not train our model end-to-end, it can also be viewed as an image enhancement method. We therefore compare it to two recent image enhancement works that focus on perceptual quality. While it is difficult to achieve perfect configuration parity, both works enhance a version of VTM and report scores on a CLIC dataset. We take DIRAC-100 trained on a hyperprior base codec and apply it on top of VTM-17.1 without finetuning. Our scores are calculated on the same dataset that each method reports on

**Kim et al. (2020)** train a model on the CLIC2020 train dataset combined with MS COCO (Lin et al., 2014) to enhance VTM-7.3. They report scores on the CLIC2020 test set, and use the *Perceptual Index (PI)* as a perceptual metric, which is defined as:

$$PI = \frac{(10 - Ma) + NIQE}{2} \ . \tag{15}$$

*Ma* is a no-reference quality metric defined in Ma et al. (2017), going from 0 (poorest quality) to 10 (best quality).

**Wang et al. (2022)** base their approach on the *Enhancement Compression Model (ECM)*[2], which in turn is built on VTM. They train a model to enhance its output in an in-loop fashion, meaning it's not a post-processing approach. As a recent enhancement approach focused on perceptual quality, we still decided to include it. They train on DIV2K (Agustsson & Timofte, 2017) and report scores on the CLIC2022 validation set, using LPIPS as a perceptual metric.

Even though we apply our model on VTM without finetuning, we achieve performance comparable to Wang et al. (2022) with similar perceptual quality improvement, at the cost of more PSNR. We assume that with additional finetuning, our approach would outperform theirs.

## C.3 PERCEPTUAL METRICS

In Fig. 10 we show additional metrics, namely the Inception Score (IS) (Salimans et al., 2016), Kernel Inception Distance (KID) (Bińkowski et al., 2022) and NIQE (Mittal et al., 2013). IS and KID are distribution metrics and thus not very reliable on Kodak, which only contains 24 images. As

---

[2]https://vcgit.hhi.fraunhofer.de/ecm/ECM.git

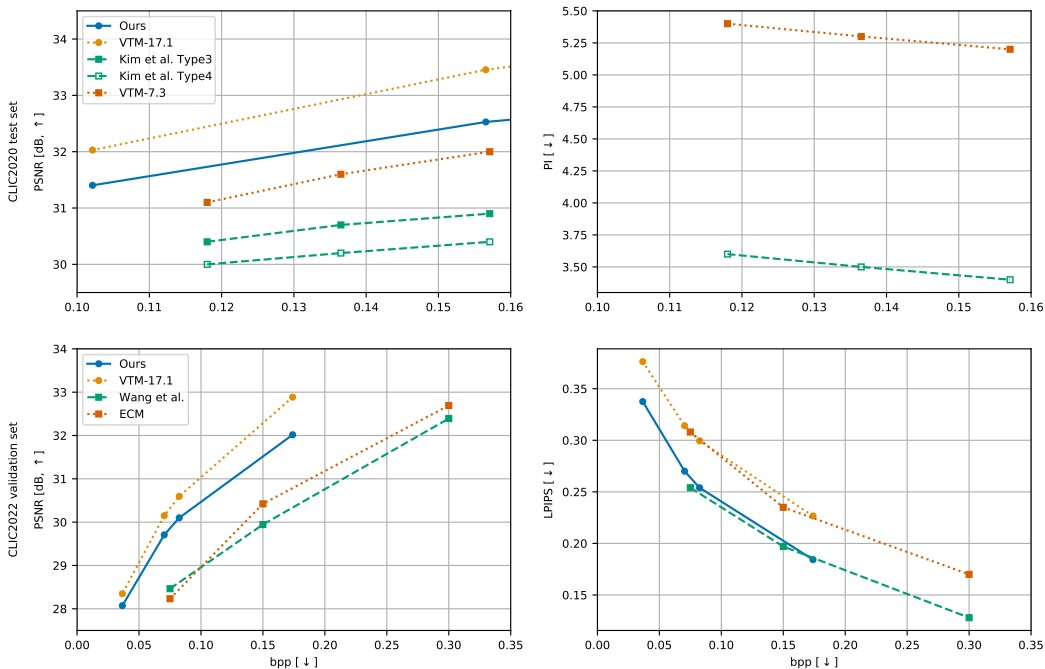

Figure 9: We compare DIRAC-100 with a VTM-17.1 base codec (trained on a hyperprior base codec, without finetuning) with two different methods, namely the work by Kim et al. (2020) (top row) and the *ArabicaPerceptual* method from Wang et al. (2022) (bottom row). The former reports PI (see Eq. (15)) as a perceptual metric on CLIC2020 test set, while the latter reports LPIPS on the CLIC2022 validation dataset. Note that PI computation is very expensive, and results for our model will be added as soon as available.

a result, we observe negative values for HifiC and our model, because we use an unbiased estimator. While overall the relative performance is similar to that seen for FID (see Fig. 4), we were surprised to see unstable scores for HiFiC in NIQE. All other methods, including ours, show a smooth decay with rate increase.

## C.4 QUALITATIVE RESULTS

We include more qualitative results below, including additional baselines JPEG and VTM. Distributed over Fig. 11, Fig. 12, Fig. 13 we show 9 images from the CLIC 2020 test set, with selected crops to zoom on salient regions.

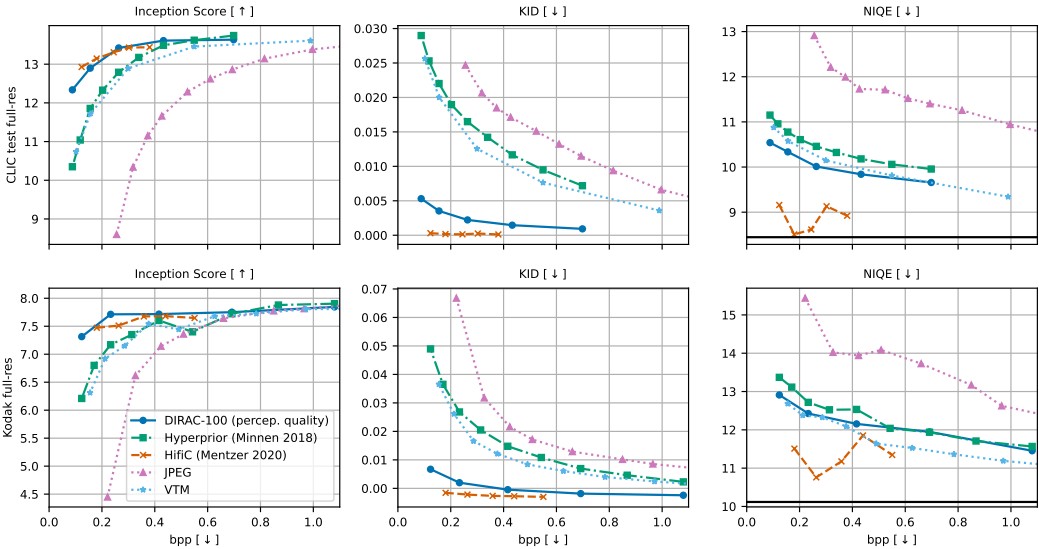

Figure 10: Additional perceptual metrics for CLIC2020 test set (top) and Kodak dataset (bottom). Note that distribution metrics (Inception Score and KID) are not reliable for small datasets like Kodak. Solid black lines show original data scores for no-reference metric NIQE.

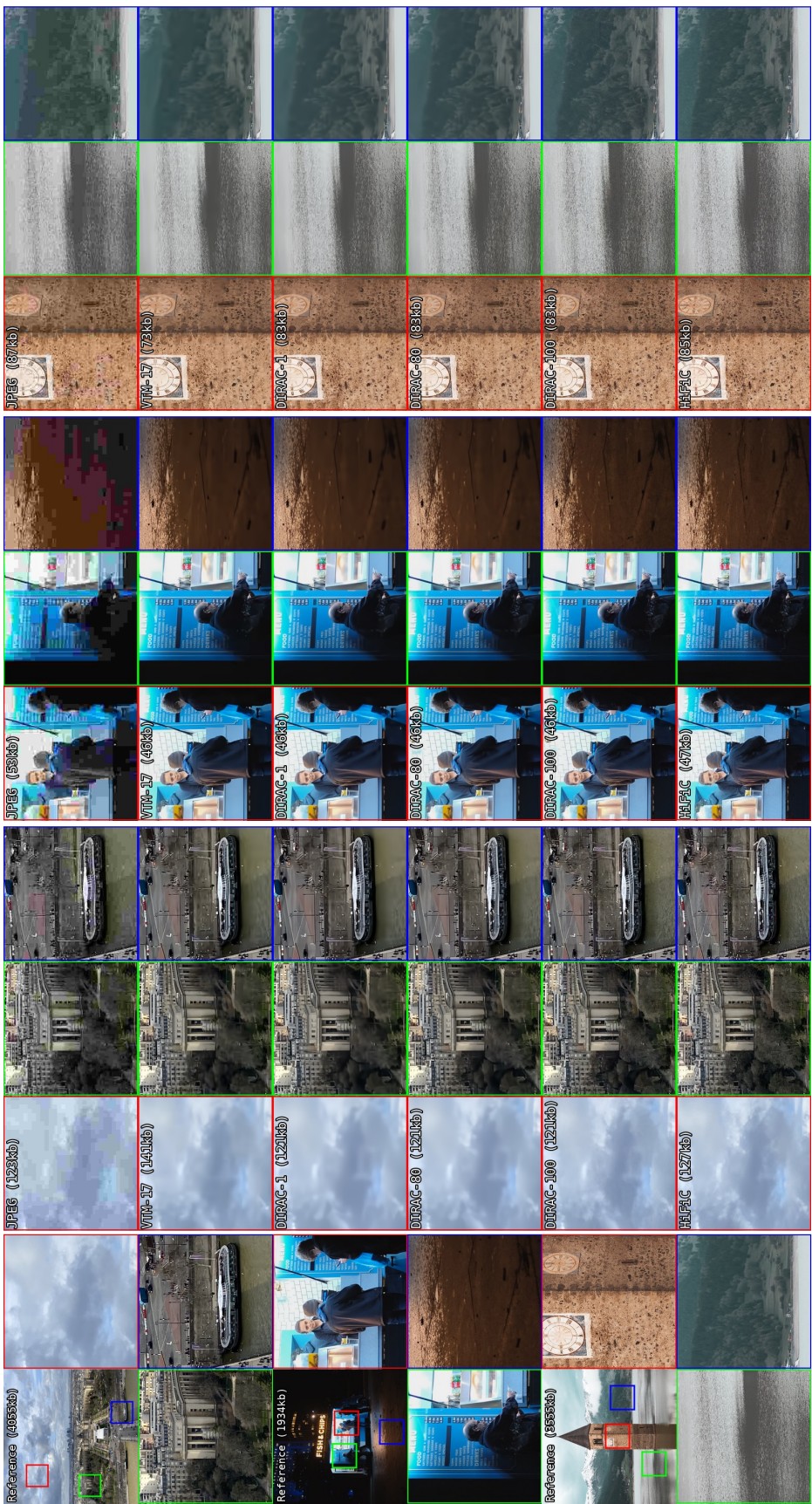

Figure 11: Qualitative examples (part 1). Left panel shows reference, then within each panel each row shows zooms for JPEG, VTM, DIRAC at sampling step 1, 80 and 100, and finally HiFIC. All methods are roughly at the same file size. Best viewed electronically.

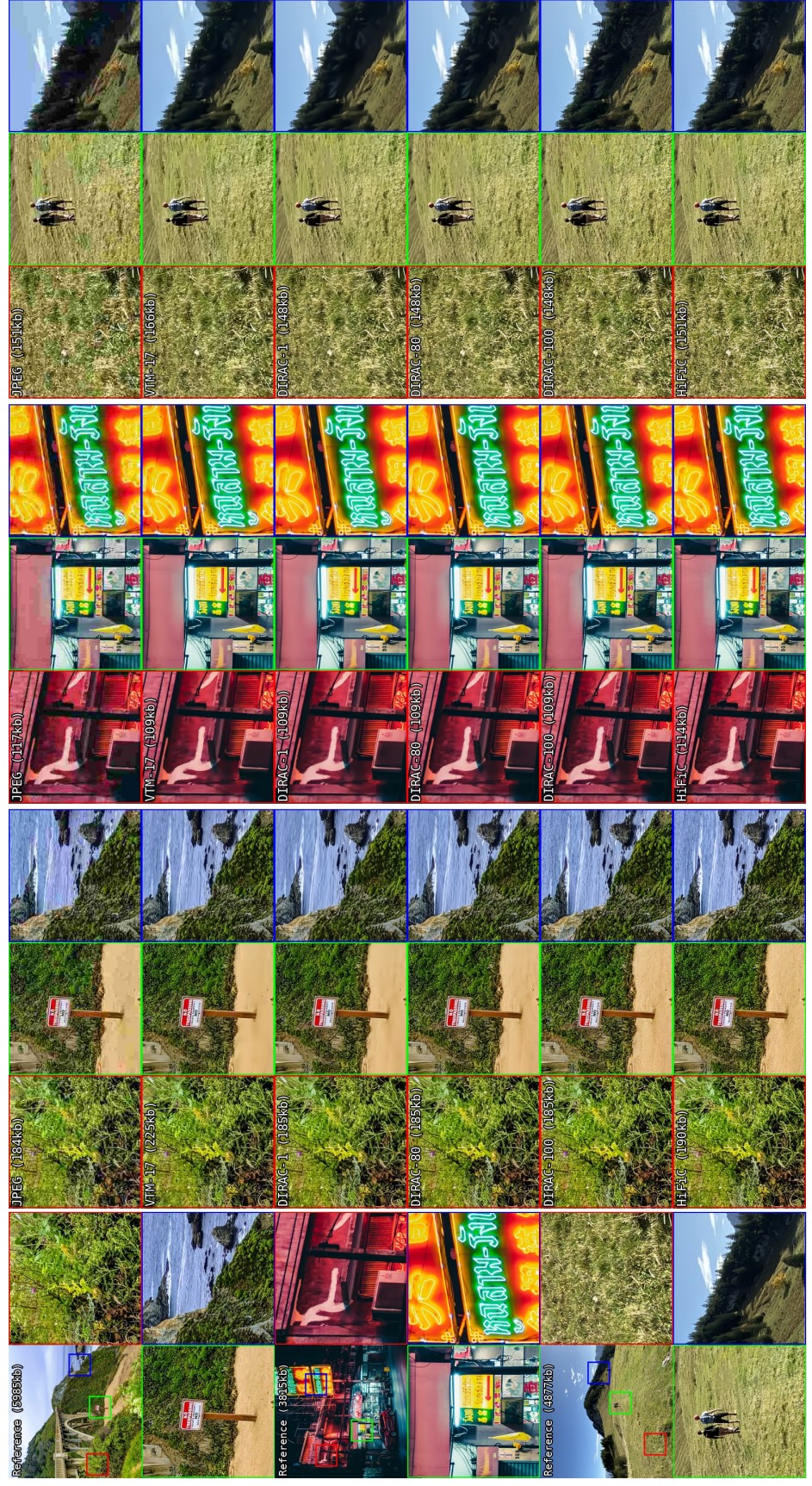

Figure 12: Qualitative examples (part 2). Left panel shows reference, then within each panel each row shows zooms for JPEG, VTM, DIRAC at sampling step 1, 80 and 100, and finally HiFIC. All methods are roughly at the same file size. Best viewed electronically.

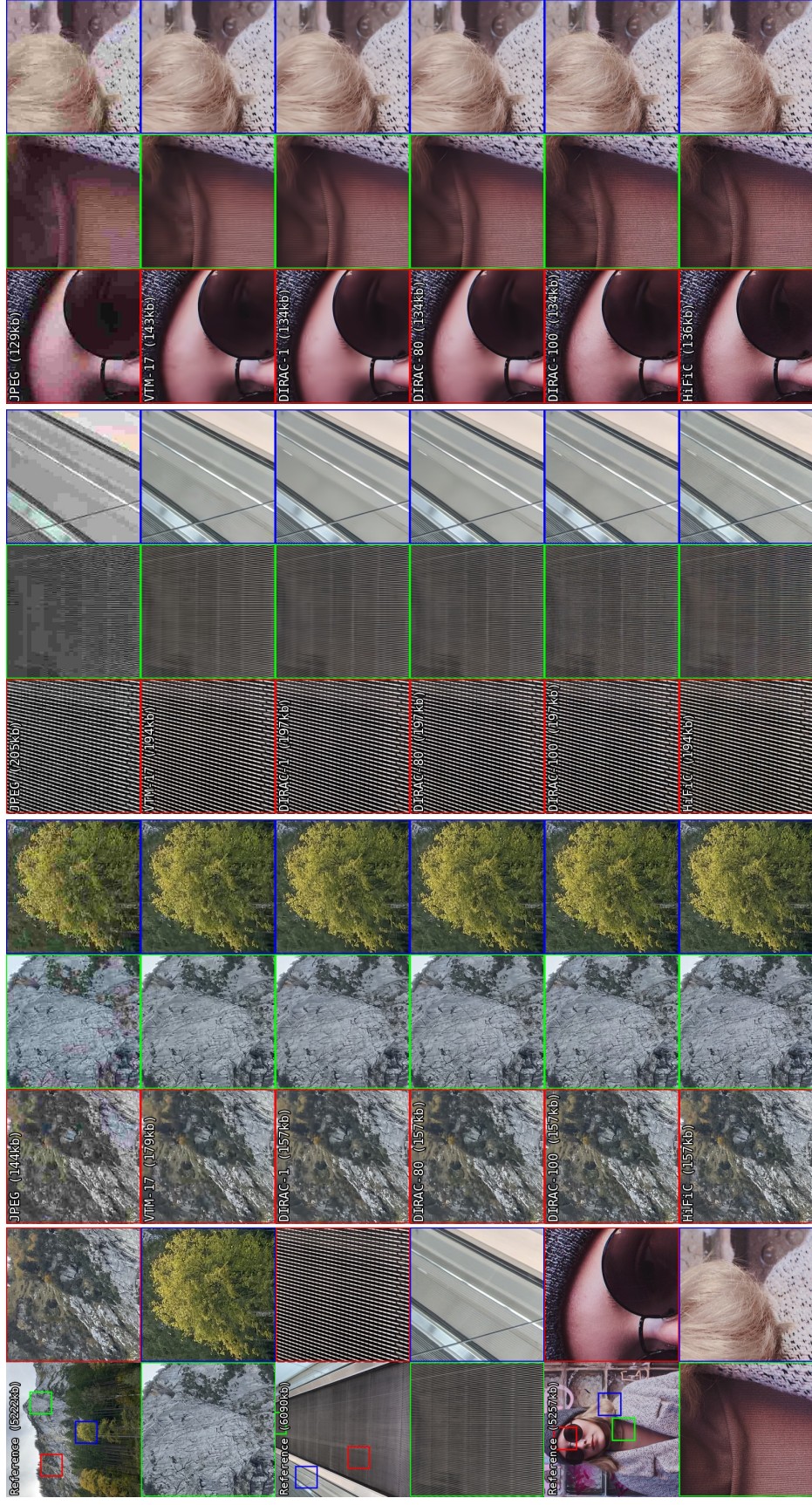

Figure 13: Qualitative examples (part 3). Left panel shows reference, then within each panel each row shows zooms for JPEG, VTM, DIRAC at sampling step 1, 80 and 100, and finally HiFIC. All methods are roughly at the same file size. Best viewed electronically.

