# OpenReview forum: "Neural Image Compression with a Diffusion-based Decoder"
_ICLR.cc/2023/Conference — Submitted to ICLR 2023_

### Official Review · Reviewer_G974 · 2022-10-24

**Confidence:** 5
**Correctness:** 4
**Technical Novelty And Significance:** 2
**Empirical Novelty And Significance:** 2
**Recommendation:** 3

**Clarity, Quality, Novelty And Reproducibility:**

The clarity of this work is clear, and it offers a diffusion-based solution for post-processing. It should be easy for others to reproduce.

**Strength And Weaknesses:**

Strengths
1. This work enables users to choose if an image is decoded with high fidelity or high perceptual quality by changing the sampling steps of the diffusion-based residual augmentation module.
2. This work shows that in the compression setting, the number of sampling steps can be reduced dramatically, although typical diffusion probabilistic models are notoriously expensive to sample from.

Weaknesses
1. This work belongs to the post-processing category, and the correlation with compression itself is not convincing and attractive to some extent. That is to say, the diffusion method, which has been illustrated as effective ways in many other low-level tasks (e.g., restoration, super-resolution) can be easily migrated in this compression task. As a comparison, the work HiFiC is an end-to-end trained method using generative loss to improve perceptual quality. The novelty of this work, especially for the specific compression task is quite limited.
2. The curve diagrams that are appeared in the paper are not very intuitive for understanding, especially Fig.5. The author should modify them.

**Summary Of The Paper:**

This paper proposes a post-processing method for neural image compression using a diffusion probabilistic model to improve the perceptual quality of the reconstruction results . With a variable-rate model that has been fully trained, no more bits need to be coded. The fidelity and perceptual quality are determined by adding residuals to different degrees with controllable sampling steps.

**Summary Of The Review:**

The paper proposes a good choice for compressed image enhancement. Further studies should be conducted to improve the novelty.

---

> ### Author Response · Authors · 2022-11-16
> **Response to Reviewer G974**
>
> Thank you for the feedback!.
>
> We acknowledge that in its current form, the method follows a compress-then-enhance scheme, meaning it is closely related to postprocessing.
> To recognize this, we are adding different base codecs and comparisons to postprocessing approaches in the updated Appendix.
>
> That said, the overall architecture is a neural codec.
> Pretraining or training in stages is a common strategy, and we adopt it here as well.
> For example, HiFiC [Mentzer 2020] is trained end-to-end, but starts with a pretraining stage of 1M iterations where no GAN loss is used.
> Latent diffusion [Rombach 2022] pretrains the encoder/decoder, then freezes these before training a diffusion model.
>
> The work of [Preechakul 2021] shows that training end-to-end with a diffusion-based decoder is feasible, but doing so would complicate the training setup, because it requires tuning and additional loss factor ($\lambda_{perception}$ in eq. (5)). It further doesn't guarantee that the sampling procedure starts from a point of maximum fidelity.
> As we did not see improvement when training or finetuning with the rate-distortion-perception loss, we kept the stages separate for simplicity.
> It is possible that there are practical advantages to, for example, conditioning the diffusion model on the transmitted latent variable, and training end to end.
> However, in theory, the latent should not contain more information than the reconstruction.
> An approach that is based on latent conditioning and end-to-end training is presented by [Yand \& Mandt 2022], and our model outperforms theirs by a large margin.
>
> Finally, we are improving the descriptions of our figures. For figure 5 specifically, we're removing the yellow curve (high rate model) and the dashed line for the one-shot prediction. In our opinion, this makes it much easier to see that skipping the first 80 sampling steps doesn't hurt performance and can even improve it. It would be helpful if you could point out what other modifications you are looking for.
>
> [Mentzer 2020]
> "High-Fidelity Generative Image Compression", F. Mentzer et al., NeurIPS 2020
>
> [Rombach 2022]
> "High-Resolution Image Synthesis with Latent Diffusion Models", R. Rombach et al., CVPR 2022
>
> [Preechakul 2021]
> "Diffusion Autoencoders: Toward a Meaningful and Decodable Representation", K. Preechakul, CVPR 2022
>
> [Yang \& Mandt 2022]
> "Lossy Image Compression with Conditional Diffusion Models", R. Yang \& S. Mandt, ECCV 2022 Workshop on Uncertainty Quantification for Computer Vision

---

### Official Review · Reviewer_Yvxp · 2022-10-24

**Confidence:** 3
**Correctness:** 3
**Technical Novelty And Significance:** 1
**Empirical Novelty And Significance:** 1
**Recommendation:** 3

**Clarity, Quality, Novelty And Reproducibility:**

- For the reasons stated in the weakness above, I think the author's contributions were not clear.
- Also, it was difficult to follow where the proposed methods are located, as the sections on conventional research and proposed methods are not clearly separated.
- The motivation of tune the degree of post-processing of image quality enhancement was insufficiently explained and the value is seem to be limited.
- The background theory of diffusion model is well explained in the appendix. I found the text relatively easy to read.
- The authors did not provide source code, but they experimented with the source code of Dhariwal & Nichol (2021), and Table 1 describes the parameters. If the authors did not modify the base source code, it should be reproducible.

**Strength And Weaknesses:**

Strength
- It may be new that the diffusion process was utilized as a quality enhancement post-processing network for the compressed image, allowing the degree of post-processing can be adjusted at the test phase by early stopping the diffusion sampling.

Weakness
- The compression part used the conventional Minnen 2018, and the architecture of the diffusion process used Dhariwal & Nichol 2021 and Preechakul 2022, it seems no particular innovations. Although the authors combined these with the loss function but they did not seem to success to learning the effective entire end-to-end process, so the research contents is considered to be more like an image enhancement task with the diffusion model. (In this sense, it may be interesting to consider and verify to what extent the diffusion model is effective in reducing compression artifacts for other codecs, including traditional ones.)
- Perceptual quality was only evaluated by LPIPS and FID, and no subjective image quality evaluation was performed.
- Since this paper is considered to be close to the task of image enhancement, i think it should be quantitatively compared with other approaches of enhancement networks that aim to improve the perceptual quality of compressed images (e.g. Y. Kim et al., Towards the Perceptual Quality Enhancement of Low Bit-rate Compressed Images, CVPRW 2020).

**Summary Of The Paper:**

The authors applied the diffusion model to the decoded image of the deep image compression model (Minnen 2018) learned with MSE, aiming to improve perceptual quality. Experimental results show that post-processing with the diffusion model improve LPIPS and FID than the original model (Minnen 2018).
The authors also show that the degree of post-processing can be adjusted at the test phase by early stopping the diffusion sampling.

**Summary Of The Review:**

Although the direction of using the diffusion model to improve the visual quality of compressed images is interesting, but i think the contents of this paper need to be reconsidered because of the difficulty in seeing the important contributions of the proposed method, the lack of sufficient evaluation, and the insufficient comparison with related studies.

---

> ### Author Response · Authors · 2022-11-16
> **Response to Reviewer Yvxp**
>
> Thank you for your feedback on how to improve our paper.
>
> As you point out, our main contribution is indeed the novel tradeoff of distortion and perception, which can be adjusted dynamically at test time on the receiver side.
> We are not claiming any novelty in the context of conventional neural codecs, or in the space of diffusion models in general.
> We tried to clarify this using separate methods sections that first introduce diffusion models (2.1), then neural codecs (2.2), and only then our own contribution (2.3). We are improving the introduction to 2.3 to make this clearer.
> While simply combining a base codec with a diffusion model does not constitute significant novelty, we believe that our design choices and resulting findings (dynamic distortion-perception tradeoff and a substantial reduction in sampling steps) are significant contributions and warrant publication.
>
> You point out that our work can also be viewed from the perspective of image-enhancement models, since we do not train end-to-end. This is of course true, and we thank you for this suggestion. We are adding a section to the appendix that compares to image enhancement works. We are also adding a section that applies our diffusion model on top of other base codecs (including standard ones), showing that the model is robust to these changes and thus also effective at reducing different compression artifacts.
>
> Your suggestion of a user study is also very sensible, but difficult to realize in our case.
> Typical user studies compare e.g. 2 images from competing methods, and users decide which they prefer.
> The idea of our approach is that a working point on the distortion-perception curve can be selected dynamically. If we predefine the "optimal" point ourselves for each test case, we're inadvertently introducing some bias; if we allow the evaluator to tune it themselves, the comparison to other methods is no longer blind.
> However, to improve the evaluation of perceptual quality, we're adding a section with more perceptual metrics (namely IS, KID and NIQE) in the appendix.
>
> Finally, we would like to ask clarification on your correctness assessment, where you say "Several of the paper’s claims are incorrect or not well-supported". We believe that we have presented all necessary derivations and experiments to support our claims, so it would be helpful if you could point out what specifically is missing in this regard.

---

> > ### Comment · Reviewer_Yvxp · 2022-12-04
> > **Thank you for response**
> >
> > First, I would like to thank the authors for the revision in such a short period.
> >
> > After looking at the revised paper, because of the additional evaluation of several additional image quality metrics, although not subjective evaluation, I think the description was slightly improved, so I gave it a slightly better rate of Correctness.
> >
> > However, I believe that there are still challenges to publishing this paper. The main objective of this paper still does not seem to be clear.
> > 1) Is the main objective to improve perceptual image quality (PERFORMANCE) with a new approach? If this case, I think the comparison and analysis with other enhancement approaches and novelty as a method should be enhanced. The evaluation of figure 8 is interesting, but the authors seem not able to compare it with conventional enhancement networks under the same conditions, so the superiority of the proposed method is unknown. It needs a clear explanation of where the potential for superior performance lies with other enhancement methods (e.g. GANs).
> > 2) Is the main objective to enable the smooth adjustment of perceptual quality and PSNR (FUNCTIONALITY)?
> > If this case, the significance and value of the functionality should be carefully explained. And I think a more careful analysis of the benefits of adjusting image quality by the number of diffusion samplings is needed. Because it seems easy to adjust the degree of blending residuals output from the enhancement network at the test phase as a naive approach, and it may need to show the advantage of using diffusion processes.
> >
> > It seems to me that these are different axes of objective (value), and asserted both at same time might makes the purpose of the paper's content unclear. Since the perspectives of this paper are interesting, I thought it would be better to focus on one claim and enhance the method, evaluation, and analysis.
> >
> > Clarifying these might require a major change in the structure of the paper, so I did not change my rating.

---

> > > ### Author Response · Authors · 2022-12-13
> > > **Thank you for the additional feedback**
> > >
> > > We are glad that you find the additional evaluations useful. It is helpful to know that it was not clear whether we were aiming to improve perceptual quality over the state of the art, or to demonstrate a methodological improvement. Our goal is the latter, and we shall make it more clear in a future revision.
> > >
> > > We think that being able to dynamically tradeoff fidelity and perceptual quality at a given rate, and purely on the receiver-side, is a useful new concept, as evidenced by the examples we show (e.g. figures 1 & 3). Simultaneously achieving state-of-the-art FID and PSNR would have been surprising, as more flexibility usually comes at the cost of some performance.
> > >
> > > In the main text, we implement DIRAC with a simple base codec i.e., mean-scale hyperprior. Additionally, in the revised document, we show that it is possible to swap out the base codec for a slower and more powerful variant. As a result, any base codec with state-of-the-art fidelity should allow our model to achieve this receiver-side distortion-perception tradeoff.

---

### Official Review · Reviewer_gzCy · 2022-10-25

**Confidence:** 3
**Correctness:** 4
**Technical Novelty And Significance:** 2
**Empirical Novelty And Significance:** 3
**Recommendation:** 6

**Clarity, Quality, Novelty And Reproducibility:**

Overall, the paper was clear and of high quality.  It took a while to understand the graph in Figure 1, but it is showing a lot of information and it eventually made sense.

I don't see any major hurdles to reproducibility.

Novelty is mild. On the one hand, I haven't seen a paper address the full rate-distortion-perception space with a single model before, I think that's quite interesting. On the other hand, using diffusion for image compression was bound to happen (and the authors note concurrent research). It may be that diffusion marks a significant step forward for the neural image compression subfield, but that's not clear yet. In other words, a comparison with a residual GAN-based model (with a multirate base codec) would also allow targeting different points in the rate-distortion-perception space and would probably have a lower runtime. Would it yield better perceptual results? I don't know, though the results in this paper show that HiFiC (which is GAN-based by not multirate or flexible in terms of distortion vs. perception) does give better perceptual quality.

**Strength And Weaknesses:**

The primary strength of the paper is a the creation of a single model that can flexibly target different points in rate-distortion-perception space. The use of a residual diffusion model allows the model to target perceptual quality. Diffusion models can be slow (especially in pixel-space) but the authors mitigate this downside to some extent by showing that a large number of diffusion steps can be skipped with minimal degradation to perceptual quality (specifically, they show that up to 80 steps can be skipped when evaluating up to 100 diffusion steps).

There are two primary weaknesses of the paper. First, as mentioned above, the decoder needs to run a diffusion model (up to 20 steps), which means it's quite a bit slower than a typical neural image compression model (and these are already an order of magnitude slower than standard codecs). Second, even after 100 diffusion steps, the perceptual quality does not match/exceed the performance of HiFiC, a GAN-based image compression model that targets perceptual quality.

To some extent, the same argument can be made about distortion, i.e. DIRAC-1, which minimizes distortion (max PSNR) is not SOTA. However, since the approach uses two separate models, it seems fair to say that any codec could be plugged in to the first stage (which generates a low-distortion image) and thus non-SOTA PSNR is not a deficiency of the approach per se.

Finally, a user study is always appreciated when perceptual quality is a goal. So that would strengthen the paper, but I don't think publication should require it so long as reasonable perceptual metrics (like FID and LPIPS which are reported) are used.

**Summary Of The Paper:**

This paper uses a residual diffusion-based model to augment a learned image compression model. The combined model has two main advantages. First, the diffusion post-processing improves perceptual quality (at the expense of increased distortion, i.e. FID improves while PSNR falls). Second, the approach leads to a single model that can target different different distortion-perception trade-offs at inference time.

The model is also multirate meaning that a single model can target different bit rates. This capability alone is not novel, but the end result here is a single compression model that can be adjusted at runtime to target different points in the 3D rate-distortion-perception space. As far as I know (and as the authors claim) this is the first model to have this flexibility / capability.

The final contribution of the paper is showing that the model can skip a large percent of the diffusion steps with only a minor reduction in perceptual quality (as measured by FID). It's not entirely clear *why* this works, but the empirical demonstration is convincing, and the authors argue that it implies that the noise schedule is suboptimal.





**Summary Of The Review:**

I have a "marginal accept" score because a flexible model in terms of rate, distortion, and perceptual is very interesting, and I haven't seen that before. Using diffusion for image compression makes sense but isn't particularly novel on its own. As mentioned above, further comparison to GAN-based methods and via a user study would strengthen the paper as would outperforming HiFiC, the generative image compression baseline.

---

> ### Author Response · Authors · 2022-11-16
> **Response to Reviewer gzCy**
>
> Thank you for your positive review!
>
> We agree with the two main weaknesses you point out.
> As one of the first works to tackle image compression with diffusion models, we make a concrete contribution: we reduce computational complexity of sampling, and obtain competitive distortion-perception performance.
> At the same time, solving both issues simultaneously is challenging.
>
> 1) All diffusion models require at least a few sampling steps, and needing at most 20 steps
> (and often far fewer, depending on the desired operating tradeoff)
> makes our procedure one of the most efficient ones.
> It is common to see papers using $>$100 sampling steps to achieve the best possible (perceptual) results (as an example, the concurrent work by [Yang \& Mandt 2022] uses 500 steps).
> There are ways for us to improve sampling efficiency, e.g. by improving the noise schedule [Kingma 2021], using more advanced multi-step or higher-order samplers [Liu 2022, Jolicoeur-Martineau 2021], and distilling the model [Salimans \& Ho 2022].
> Our main point here was that our construction (base codec + modeling residuals) makes sampling efficient, orthogonal to the aforementioned speedups.
> We believe this is a significant finding.
>
> 2) Indeed, we did not achieve the state-of-the-art performance in distortion or perception, but propose a model that performs competitively along both axes.
> The main advantage of the approach is the convenient test-time tradeoff between these two axes, as both axes can be important depending on use case and user requirements/preferences.
> As suggested, we show in the updated Appendix that our setup allows swapping to stronger base codecs, for instance the standard codec VVC.
> This works without finetuning, indicating the model is robust to the choice of base codec.
> Using VVC as base codec would be slower than the mean-scale hyperprior, but provided we use a diffusion model anyway, one might argue time complexity is not our main issue.
>
> A multi-bitrate residual GAN-based approach is certainly an interesting idea as well, and would have been a useful baseline.
> However, most GAN-based methods conflate distortion and perception with no easy way to navigate this tradeoff.
> In contrast, we show that diffusion models provide a user-friendly way to trade off these two axes dynamically. It is not obvious how to achieve the same with a GAN-based approach.
>
> You seem to agree that the test-time receiver-side tunability of distortion vs. perception is a useful and practical contribution, once diffusion models themselves become more practical.
> Improving the efficiency of diffusion models is a major research direction, and one that we do make a significant contribution to in the context of neural codecs.
> We are also adding a section on using different base codecs, which demonstrates that our model is robust to this change and does indeed achieve higher PSNR.
> In light of the above, we hope that you're willing to increase your rating to "Accept".
>
>
>
> [Yang \& Mandt 2022]
> "Lossy Image Compression with Conditional Diffusion Models", R. Yang \& S. Mandt, ECCV 2022 Workshop on Uncertainty Quantification for Computer Vision
>
> [Kingma 2021]
> "Variational Diffusion Models", D. Kingma et al., NeurIPS 2021
>
> [Liu 2022]
> "Pseudo Numerical Methods for Diffusion Models on Manifolds", L. Liu et al., ICLR 2022
>
> [Jolicoeur-Martineau 2021]
> "Gotta Go Fast When Generating Data with Score-Based Models", A. Jolicoeur-Martineau et al., arXiv 2021
>
> [Salimans \& Ho 2022]
> "Progressive Distillation for Fast Sampling of Diffusion Models", T. Salimans \& J. Ho, ICLR 2022

---

### Comment · Area_Chair_AwVK · 2022-11-18
**Responses**

Dear Reviewers,

Do you have any comments/replies to author's responses - it would be great if you could respond to them. Have they changed your opinion on the paper?

Kind regards,
AC

---

### Decision · Program_Chairs · 2023-01-20

**Decision:**

Reject

**Justification For Why Not Higher Score:**

- The method is closer to image enhancement methods and should be better compared to those. In this sense it is also not end to end compression method and perhaps could be made so.
- Better explanation of the benefit of adjusting image quality by number of diffusion steps.
- Overall the method is not very novel
Other potential improvements
- Subjective evaluation
- Comparison to using GAN instead of diffusion model in the same setting.


**Justification For Why Not Lower Score:**

N/A

**Metareview: Summary, Strengths And Weaknesses:**

The method adds a diffusion model to improve images that are compressed using auto-encoding system and is able to traverse rate-distortion-perception tradeoff during test time.
The drawbacks/potential improvements of the paper are the following (from reviewer responses).
- The method is closer to image enhancement methods and should be better compared to those. In this sense it is also not end to end compression method and perhaps could be made so.
- Better explanation of the benefit of adjusting image quality by number of diffusion steps.
- Overall the method is not very novel
Other potential improvements
- Subjective evaluation
- Comparison to using GAN instead of diffusion model in the same setting.